# EarthMind: Leveraging Cross-Sensor Data for Advanced Earth Observation Interpretation with a Unified Multimodal LLM

## Abstract

Earth Observation (EO) data analysis is vital for monitoring environmental and human dynamics. Recent Multimodal Large Language Models (MLLMs) show potential in EO understanding but remain restricted to single-sensor inputs, overlooking the complementarity across heterogeneous modalities. We propose EarthMind, *a unified vision-language framework* that handles both *single- and cross-sensor* inputs via an innovative hierarchical cross-modal attention (*i.e.*, HCA) design. Specifically, HCA hierarchically captures visual relationships across sensors and aligns them with language queries, enabling adaptive fusion of optical and Synthetic Aperture Radar (SAR) features. To support cross-sensor learning, we curate *FusionEO*, a 30K-pair dataset with diverse annotations, and establish *EarthMind-Bench*, a 2,841-pair benchmark with expert annotations for perception and reasoning tasks. Extensive experiments show that EarthMind achieves state-of-the-art results on EarthMind-Bench and surpasses existing MLLMs on multiple EO benchmarks.

## 1 Introduction

Multimodal Large Language Models (MLLMs) have revolutionized vision-language understanding by integrating powerful language models (OpenAI, 2023; Team et al., 2023) with visual encoders, achieving remarkable performance across diverse tasks, including image captioning (Xu et al., 2024; Chen et al., 2024a), visual question answering (Li et al., 2023; Liu et al., 2023a; Li et al., 2024a), and visual grounding (Lai et al., 2024; Rasheed et al., 2024; Yuan et al., 2025). Earth Observation (EO) represents a particularly critical application domain with far-reaching implications for environmental monitoring (Wójtowicz et al., 2016) and disaster response (Van Westen, 2000). However, the distinctive characteristics of EO imagery create a substantial domain gap that limits the effectiveness of MLLMs trained on natural images. Recent efforts have addressed this challenge by developing EO-specific instruction tuning datasets (Kuckreja et al., 2024; Muhtar et al., 2024; Zhan et al., 2025; Zhang et al., 2024b; Hu et al., 2025) and specialized architectures, demonstrating significant improvements in adapting MLLMs to remote sensing applications.

Despite these advances, existing MLLMs for EO remain *fundamentally limited by their reliance on single-sensor inputs*, failing to exploit the complementary nature of heterogeneous sensing modalities (Fuller et al., 2023; Liu et al., 2024a). Modern EO platforms routinely provide synchronized multimodal data: optical sensors like Sentinel-2 capture high-resolution multispectral imagery rich in spectral and textural information, while SAR sensors like Sentinel-1 deliver all-weather, day-night observation capability. These modalities exhibit natural complementarity—optical sensors excel in clear conditions with detailed spectral signatures but suffer from cloud occlusion and illumination dependency, whereas SAR penetrates atmospheric interference but exhibits lower signal-to-noise ratios and speckle noise. As illustrated in Fig. 1, this fundamental trade-off underscores that *no single sensor can comprehensively capture scene information*, necessitating advanced and effective cross-sensor fusion strategies for robust and comprehensive EO data understanding.

However, achieving effective cross-sensor fusion presents challenges. The fundamental heterogeneity between optical and SAR data, rooted in distinct imaging principles and signal characteristics, makes naive approaches like token concatenation ineffective. While traditional fusion methods (Zhang

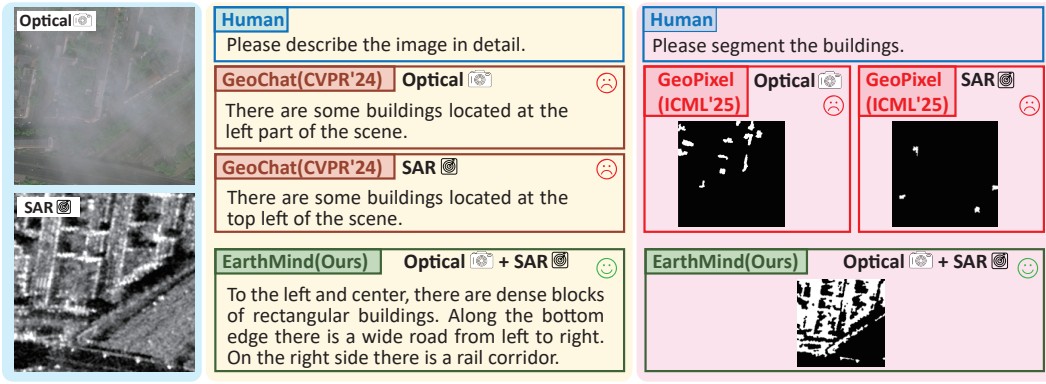

Figure 1: From a single Optical or SAR image, GeoChat (Kuckreja et al., 2024) and GeoPixel (Shabbir et al., 2025) give inaccurate VQA or segmentation results. EarthMind first utilize cross-sensor information to boost several tasks.

et al., 2025a; Schmitt et al., 2017; Li et al., 2022) show promise in specific contexts, they remain constrained by *task-specific architectures* and *lack the flexibility for diverse vision-language tasks*.

To address these limitations, we present EarthMind, a unified MLLM tailored for EO that seamlessly integrates both single-sensor and cross-sensor data. This integration enhances representational capacity, enabling the model to couple global scene context with local spatial detail and thereby supporting tasks ranging from high-level reasoning to fine-grained pixel-level grounding. At the heart of EarthMind is the *hierarchical cross-modal attention (*i.e.*, HCA)* design, a lightweight mechanism that equips MLLMs with adaptive cross-sensor fusion. By learning spatially adaptive weights tied to sensor density and query relevance, HCA mitigates modality imbalance and preserves complementary SAR cues otherwise suppressed by optical bias. To overcome the scarcity of paired cross-sensor data, we further introduce *FusionEO*, a 30K-sample instruction-tuning dataset generated through an automated pipeline, ensuring the semantic diversity and task coverage required for effective training.

Additionally, we introduce EarthMind-Bench, a large-scale benchmark for evaluating cross-sensor MLLMs in EO scenarios. It enables flexible evaluation with both single- and multi-sensor inputs, spans tasks from *image-level understanding* to *pixel-level segmentation*, and covers reasoning from perception to high-level analysis. Comprising 2,841 expertly annotated pairs, EarthMind-Bench provides a unified platform to rigorously assess MLLM capabilities in interpreting and reasoning over EO data. Overall, the effectiveness of EarthMind is demonstrated in two key aspects:

- It successfully exploits cross-sensor complementarity to enhance diverse EO understanding tasks—including dense captioning, VQA, and segmentation—achieving state-of-the-art performance on our multi-sensor EarthMind-Bench with only 4B parameters.

- It also demonstrates superior performance on single-sensor benchmarks across various tasks, spanning image-level classification and VQA, region-level visual grounding, and pixel-level referring expression segmentation.

## 2    RELATED WORK

**Earth Observation MLLMs.** Building upon the success of general image MLLMs, numerous works (Hu et al., 2025; Kuckreja et al., 2024; Zhan et al., 2025; Zhang et al., 2024b;a; Muhtar et al., 2024; Luo et al., 2024; Shabbir et al., 2025; Soni et al., 2024) have attempted to transfer these capabilities to the EO domain. A key challenge is the scarcity of instruction-tuned EO datasets. RSGPT (Hu et al., 2025) addressed this by proposing the first large-scale EO image-text paired dataset, enabling conversational tasks such as image captioning and VQA. GeoChat (Kuckreja et al., 2024) and SkyEyeGPT (Zhan et al., 2025) extended capabilities to region-level visual grounding through region-centric instruction data. LHRS-Bot (Muhtar et al., 2024) leverages large-scale EO imagery aligned with OpenStreetMap annotations to improve pretraining. To enhance complex reasoning, SkysenseGPT (Luo et al., 2024) introduces the FIT-RS dataset focusing on spatial entity relationships. GeoPixel (Shabbir et al., 2025) pushes the boundary to pixel-level grounding through

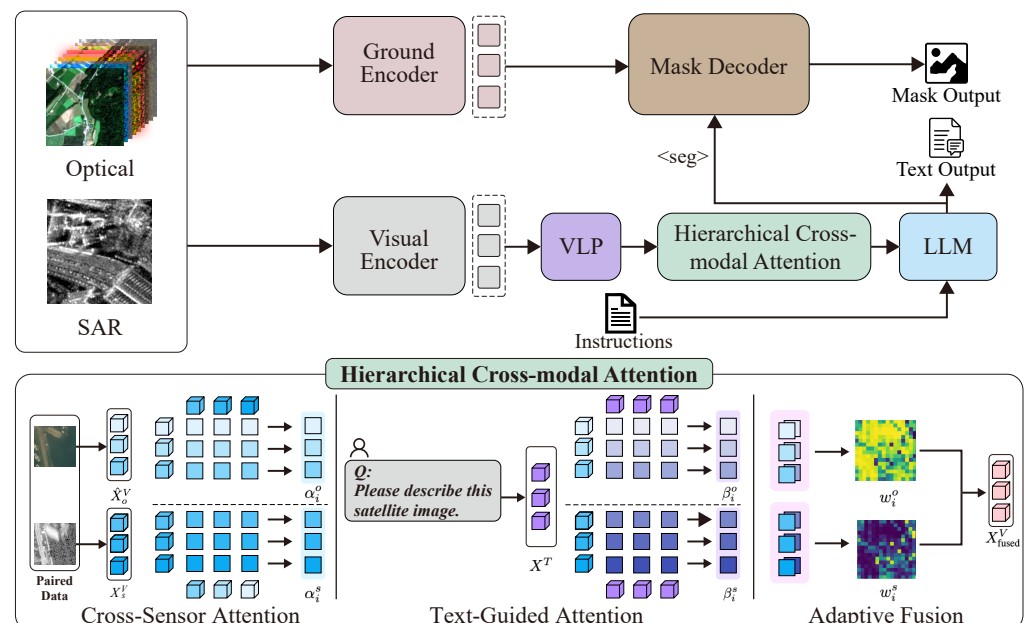

Figure 2: An overview of EarthMind, which supports both single-sensor and multi-sensor EO data understanding, including high-level image reasoning and pixel-level grounding tasks. The framework facilitates cross-sensor fusion through Hierarchical Cross-modal Attention (HCA), which selectively combines complementary features from optical and SAR data guided by both cross-modal visual relationships and input query relevance.

an automated pipeline for generating grounded conversations. Beyond optical data, EarthDial (Soni et al., 2024) incorporates diverse modalities including multispectral, hyperspectral, and SAR data to improve generalization. Despite these advances, current MLLMs remain limited in leveraging multi-sensor inputs for comprehensive EO understanding.

**Earth Observation Multimodal Benchmarks.** The rapid development of EO MLLMs has stimulated dedicated evaluation benchmarks. RSIEval (Hu et al., 2025) provides human-annotated captions and VQA pairs for remote sensing evaluation. LHRS-Bench (Muhtar et al., 2024) introduces hierarchical taxonomies for multi-dimensional assessment. VLEO-Bench (Zhang & Wang, 2024) focuses on real-world applications including urban monitoring and disaster relief. Beyond conversational tasks, VRSBench (Li et al., 2024b) and GeoChat-Bench (Kuckreja et al., 2024) incorporate grounding tasks for localization evaluation. HnstD (Pang et al., 2025) targets hallucination detection, while FIT-RSRC (Luo et al., 2024) evaluates object relationship reasoning. XLRS-Bench (Wang et al., 2025) leverages ultra-high-resolution imagery, and GEOBench-VLM (Danish et al., 2024) proposes a multi-task geospatial benchmark. Despite these advances, none explicitly assess MLLM performance under multi-sensor scenarios, where complementary cues across heterogeneous modalities are critical for robust EO understanding.

## 3 METHOD

From dense urban mapping to natural terrain analysis, EO applications require models that can natively fuse heterogeneous sensors for reliable scene understanding and decision-making. To this end, we present EarthMind, a unified vision-language framework that accepts either single-sensor or paired multi-sensor inputs and produces task-appropriate outputs, including natural-language responses and pixel-accurate masks. Sec. 3.1 outlines the overall architecture, which consists of multiple visual encoders, a vision-language projector, an LLM, and a lightweight mask decoder. Sec. 3.2 introduces our core multi-sensor fusion module, Hierarchical Cross-modal Attention (HCA). Sec. 3.3 describes the paired cross-sensor instruction-tuning corpus FusionEO and training objectives.

## 3.1 EARTHMIND ARCHITECTURE

Fig. 2 illustrates the architecture of EarthMind. Our model employs a visual encoder $\mathbf{E}_v$ for global semantic perception and a grounding encoder $\mathbf{E}_g$ for fine-grained spatial understanding. A vision-language projector (VLP) transforms visual features into a token sequence $X^V$ aligned with the language space. These visual tokens, combined with input text tokens, serve as input to the LLM, which generates text predictions accordingly. To enable dense prediction tasks, we introduce a special token "<SEG>" that guides the mask decoder $\mathbf{D}_m$ to produce fine-grained segmentation maps.

Our framework supports flexible visual input by treating heterogeneous EO data as video-like sequences. Inspired by recent advances in video-language models (Maaz et al., 2023; Shu et al., 2024), we adopt a unified data formatting strategy: single- and dual-channel SAR images are padded to form pseudo-RGB frames, while multispectral bands are grouped in triplets to construct multi-frame sequences. This approach mimics temporal video structure and enables shared encoder processing, allowing our model to exploit cross-frame dependencies and spectral complementarity.

## 3.2 CROSS-SENSOR FUSION

**Motivation.** Given multi-sensor visual inputs, we extract sensor-specific image tokens: $X_o^V \in \mathbb{R}^{P \times D}$ for optical and $X_s^V \in \mathbb{R}^{N \times D}$ for SAR modalities, where $D$ denotes the LLM's embedding dimension. The most straightforward fusion approach is concatenating these tokens into a joint representation $\hat{X}^V \in \mathbb{R}^{(P+N) \times D}$. However, this naive concatenation presents two critical limitations. First, the increased sequence length results in quadratic computational overhead due to self-attention mechanisms. Second, despite equal representation in the input, LLMs exhibit strong bias toward optical tokens, potentially overlooking valuable SAR information. This bias likely stems from the predominance of RGB images in pre-training data.

To quantitatively measure this modality imbalance, we introduce the *Modality Attention Score* (MAS):

$$A_\ell^m = \frac{\sum_{i \in \mathcal{I}} \sum_{j \in \mathcal{T}_m} \alpha_{i,j}^\ell}{\sum_{i \in \mathcal{I}} \sum_{k \in \mathcal{I}} \alpha_{i,k}^\ell} \tag{1}$$

where $\mathcal{I}$ denotes all visual tokens, $\mathcal{T}_m \subset \mathcal{I}$ represents tokens from modality $m$, and $\alpha_{i,j}^\ell$ denotes the self-attention weight from token $i$ to $j$ at layer $\ell$ in LLM. MAS quantifies the proportion of attention allocated to each modality.

Our empirical analysis (Sec. 5.4) reveals that standard MLLMs with concatenation fusion consistently exhibit higher $A_\ell^{\text{optical}}$ than $A_\ell^{\text{SAR}}$ across layers, confirming the modality bias. This motivates our Hierarchical Cross-modal Attention module, which explicitly models cross-sensor interactions for balanced and efficient multi-modal fusion.

**Hierarchical Cross-modal Attention.** We propose a hierarchical attention mechanism that captures both cross-sensor complementarity and text-guided relevance for effective multimodal fusion. Our approach operates in two stages: (1) bidirectional cross-modal attention between optical and SAR features, and (2) text-guided attention to align visual features with task requirements.

*Stage 1: Cross-Sensor Attention.* To ensure spatial alignment between modalities with different resolutions, we apply adaptive pooling to align optical features with SAR dimensions, yielding $\hat{X}_o^V \in \mathbb{R}^{N \times D}$. We then compute bidirectional attention maps:

$$A^{o2s} = \text{Softmax}\left(\frac{\hat{X}_o^V (X_s^V)^\top}{\sqrt{D}}\right), \quad A^{s2o} = \text{Softmax}\left(\frac{X_s^V (\hat{X}_o^V)^\top}{\sqrt{D}}\right) \tag{2}$$

where $A_{ij}^{o2s}$ captures attention from optical token $i$ to SAR token $j$, and vice versa for $A_{ij}^{s2o}$.

To derive modality-specific importance scores, we aggregate attention weights across spatial dimensions:

$$\alpha_i^o = \frac{1}{N} \sum_{j=1}^N A_{ji}^{s2o}, \quad \alpha_i^s = \frac{1}{N} \sum_{j=1}^N A_{ji}^{o2s} \tag{3}$$

Here, $\alpha_i^o$ represents how much attention the $i$-th optical token receives from all SAR tokens, indicating its importance from the SAR perspective.

*Stage 2: Text-Guided Attention.* We incorporate task-specific guidance by computing visual token relevance to the text prompt. Given text embeddings $X^T \in \mathbb{R}^{L \times D}$ aligned with visual features through the projector:

$$A^{o2t} = \text{Softmax}\left(\frac{\hat{X}_o^V (X^T)^\top}{\sqrt{D}}\right), \quad A^{s2t} = \text{Softmax}\left(\frac{X_s^V (X^T)^\top}{\sqrt{D}}\right) \tag{4}$$

We aggregate these scores to obtain text-relevance weights:

$$\beta_i^o = \frac{1}{L}\sum_{j=1}^{L} A_{ij}^{o2t}, \quad \beta_i^s = \frac{1}{L}\sum_{j=1}^{L} A_{ij}^{s2t} \tag{5}$$

*Adaptive Fusion.* The final fusion weights combine cross-modal complementarity and text relevance:

$$\gamma_i^o = \lambda \cdot \alpha_i^o + (1 - \lambda) \cdot \beta_i^o, \quad \gamma_i^s = \lambda \cdot \alpha_i^s + (1 - \lambda) \cdot \beta_i^s \tag{6}$$

where $\lambda$ is a learnable parameter balancing the two attention mechanisms. The fused representation is computed as:

$$X_{\text{fused},i}^V = w_i^o \cdot \hat{X}_{o,i}^V + w_i^s \cdot X_{s,i}^V \tag{7}$$

where $[w_i^o, w_i^s] = \text{Softmax}([\gamma_i^o, \gamma_i^s])$ are the normalized fusion weights. This produces final multi-modal tokens $X_{\text{fused}}^V \in \mathbb{R}^{N \times D}$ for LLM input, effectively reducing sequence length while preserving cross-modal information.

### 3.3 TRAINING

**Dataset Curation.** Although large-scale visual instruction tuning datasets exist for single-modality remote sensing imagery, paired optical–SAR data with rich text annotations remain scarce. To address this gap, we construct a diversified set of QA pairs FusionEO to facilitate effective training. Specifically, we design a three-stage pipeline: **(1) Metadata preparation**. We leverage dataset-specific metadata (e.g., modality, category, geographic location, and relevant attributes) to provide essential contextual information. These metadata are then converted into concise captions through rule-based processing. **(2) RoI-based summarization**. To enrich the textual context with visual grounding cues, we incorporate region-level information from segmentation annotations. Foreground objects or localized regions of interest (RoIs) are explicitly highlighted in the images, thereby reducing hallucination risks. Mask-rendered images, together with the short captions from Stage 1, are then used as inputs to GPT-4o to generate more detailed and descriptive captions. **(3) Self-instruct VQA generation**. Finally, we extend the caption data into diverse VQA samples. GPT-4o is prompted in a few-shot manner, using seed examples to guide the generation of varied and semantically rich question–answer pairs. Additional implementation details are provided in Appendix G.

**Training Objective.** EarthMind is trained via instruction tuning across diverse supervision formats, including VQA and segmentation tasks. For VQA tasks, we apply standard supervised finetuning way, minimizing the token-level cross-entropy loss over the ground-truth responses. For segmentation tasks, we use a combination of pixel-wise cross-entropy Loss and Dice Loss.

## 4 EARTHMIND-BENCH

We observe that existing EO benchmarks cannot evaluate the performance of MLLMs under multi-sensor input. To address this limitation, we propose EarthMind-Bench, a new benchmark constructed from high-quality paired data with optical (including both RGB and multispectral) and SAR images across various public datasets, including BigEarthNet-MM (Sumbul et al., 2021), OpenEarthMap-SAR (Xia et al., 2025), DFC2023 Track2 (Persello et al., 2023), WHU-OPT-SAR (Li et al., 2022), MSAW (Shermeyer et al., 2020), and MultiResSAR (Zhang et al., 2025b). We curate 2,841 samples from their test sets, and design a suite of 10 tasks spanning perception and reasoning, enabling a comprehensive evaluation of MLLMs in EO scenarios, as shown in Fig. 3.

To comprehensively evaluate MLLMs in EO scenarios, we design a suite of perception and reasoning tasks. The perception tasks assess fundamental understanding, ranging from coarse-grained problems such as scene classification, object existence detection, and hallucination detection to more challenging tasks including object counting, image captioning, and referring expression segmentation.

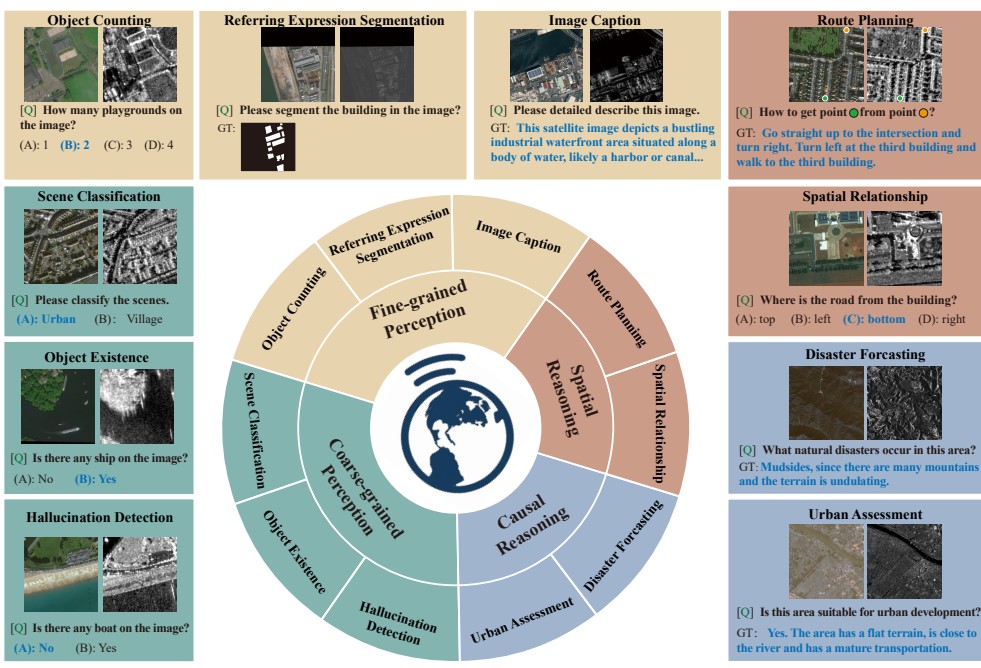

Figure 3: Examples of EarthMind-Bench. There are ten tasks to evaluate the multi-sensor fusion ability of LMMs, covering perception and reasoning levels. The LLMs are asked to solve the problem (with the ground-truth answers marked in blue) based on optical, SAR or optical-SAR fusion settings.

Building upon this foundation, the reasoning tasks are divided into spatial and causal categories. Spatial reasoning requires models to infer spatial relationships between objects or to perform route planning by generating a feasible path between a start and an end point. Causal reasoning, on the other hand, focuses on tasks such as disaster forecasting and urban development assessment, where models are expected not only to produce predictions or judgments but also to provide explanatory evidence supporting their reasoning.

For text generation tasks, we report average accuracy on multiple-choice questions and adopt GPT-based scoring to evaluate the quality of open-ended responses. For mask generation tasks, segmentation quality is measured using mean Intersection-over-Union (mIoU). All annotations are produced by human experts and further subjected to a rigorous verification process, as described in Appendix E.

## 5 EXPERIMENTS

### 5.1 IMPLEMENTATION DETAILS

EarthMind builds upon InternVL2 (Chen et al., 2024b) and SAM2 (Ravi et al., 2024), adopting a progressive learning way including three steps. First, we enhance the instruction-following capability using 1.7M general image-text data, which covers image-level captioning, VQA, region-level object understanding and text-driven segmentation. Second, we introduce 1M EO-specific multimodal data to adapt EarthMind to the remote sensing domain. Third, we utilize our synthesized multi-sensor conversation corpus, and selectively retain examples from earlier stages to mitigate catastrophic forgetting. We fine-tune EarthMind with a learning rate of 4e-5 and a batch size of 2, training only the vision-language projector, the LLM via LoRA (Hu et al., 2022), and the mask decoder. All experiments are conducted on 8 NVIDIA A100-80G GPUs. More details about training datasets and details can be seen in Appendix G.

### 5.2 BENCHMARKS

In addition to our proposed EarthMind-Bench, we evaluate EarthMind on several widely-used EO benchmarks to assess its capability across diversified multi-sensor tasks. We conduct evaluations at

Table 1: Experimental results on EarthMind-Bench, in which different evaluation settings are employed to compare multi-sensor understanding ability. "M-Avg": the average performance of multiple-choice tasks; "O-Avg": the average performance of open-ended tasks. "Referring Segmen." refers to the referring expression segmentation task, which is evaluated by mIoU. † denotes proprietary models. Bold means the best performance.

| Model | Size | M-Avg | O-Avg | Scene Class. | Object Exist. | Halluci. Detect. | Object Count. | Spatial Relation. | Referring Segmen. | Image Caption | Disaster Forecast. | Route Plann. | Urban Assess. |
|---|---|---|---|---|---|---|---|---|---|---|---|---|---|
| *Full mark* | – | 100 | 5 | 100 | 100 | 100 | 100 | 100 | 100 | 5 | 5 | 5 | 5 |
| **Evaluation on Optical modality** | | | | | | | | | | | | | |
| GPT-4o† (OpenAI, 2024) | - | **64.1** | 2.63 | 67.8 | **79.9** | **86.4** | 34.0 | **52.3** | - | **4.58** | 1.75 | **2.01** | 2.18 |
| GPT-4V† (OpenAI, 2023) | - | 55.4 | 2.12 | 60.2 | 72.9 | 75.9 | 39.0 | 29.2 | - | 3.28 | 1.54 | 1.82 | 1.86 |
| GeoChat (Kuckreja et al., 2024) | 7B | 35.5 | 1.78 | 40.9 | 51.8 | 46.8 | 18.9 | 19.0 | - | 1.92 | 1.73 | 1.33 | 2.14 |
| LHRS-bot (Muhtar et al., 2024) | 7B | 41.3 | 1.84 | 45.5 | 58.3 | 58.3 | 25.2 | 19.4 | - | 2.56 | 1.75 | 1.55 | 1.50 |
| Skysensegpt (Luo et al., 2024) | 7B | 39.8 | 1.45 | 47.4 | 56.8 | 56.7 | 26.8 | 11.1 | - | 1.68 | 1.40 | 1.18 | 1.55 |
| GeoPixel (Shabbir et al., 2025) | 7B | 53.0 | 2.06 | 55.3 | 67.8 | 73.6 | 33.5 | 34.7 | 46.8 | 2.80 | 1.80 | 1.68 | 1.95 |
| EarthDial (Soni et al., 2024) | 4B | 58.3 | 2.13 | 58.2 | 72.4 | 75.9 | 40.6 | 44.2 | - | 2.82 | 1.95 | 1.66 | 2.10 |
| **EarthMind** | 4B | 61.7 | **2.82** | 64.3 | 77.5 | 83.6 | 50.1 | 33.1 | **56.0** | 3.35 | **3.37** | **2.01** | **2.55** |
| **Evaluation on SAR modality** | | | | | | | | | | | | | |
| GPT-4o† (OpenAI, 2024) | - | 47.8 | 2.40 | 35.2 | 71.4 | 72.9 | 22.8 | 36.6 | - | 2.89 | 3.05 | 1.65 | 2.04 |
| GPT-4V† (OpenAI, 2023) | - | 44.4 | 2.22 | 30.9 | 73.3 | 67.1 | 31.0 | 19.9 | - | 2.63 | 2.98 | 1.40 | 1.85 |
| GeoChat (Kuckreja et al., 2024) | 7B | 34.3 | 1.45 | 28.6 | 49.8 | 46.8 | 27.9 | 18.5 | - | 1.78 | 1.65 | 1.25 | 1.56 |
| LHRS-bot (Muhtar et al., 2024) | 7B | 35.1 | 1.71 | 28.9 | 56.3 | 48.5 | 23.5 | 18.1 | - | 1.86 | 1.70 | 1.45 | 1.83 |
| Skysensegpt (Luo et al., 2024) | 7B | 34.2 | 1.55 | 23.5 | 53.8 | 49.2 | 33.8 | 10.7 | - | 1.76 | 1.70 | 1.35 | 1.38 |
| GeoPixel (Shabbir et al., 2025) | 7B | 44.6 | 1.80 | 35.2 | 59.0 | 65.7 | 30.5 | 32.8 | 35.9 | 2.08 | 1.97 | 1.45 | 1.68 |
| EarthDial (Soni et al., 2024) | 4B | 49.4 | 1.95 | 40.6 | 65.3 | 69.2 | 35.7 | 36.4 | - | 2.26 | 2.09 | 1.60 | 1.86 |
| **EarthMind** | 4B | 61.3 | 2.64 | 64.4 | 77.4 | 74.6 | 46.8 | 43.1 | 53.0 | 3.10 | 3.25 | 1.89 | 2.30 |
| **Evaluation on Optical-SAR Fused modality** | | | | | | | | | | | | | |
| GPT-4o† (OpenAI, 2024) | - | 61.1 | 2.28 | 64.8 | 79.6 | 86.2 | 31.6 | 43.5 | - | 3.68 | 1.59 | 1.82 | 2.03 |
| GPT-4V† (OpenAI, 2023) | - | 46.0 | 1.93 | 30.2 | 64.8 | 62.4 | 32.8 | 39.8 | - | 2.89 | 1.48 | 1.57 | 1.79 |
| **EarthMind** | 4B | 70.0 | 3.02 | 65.5 | 84.4 | 88.1 | 52.4 | 59.7 | 59.8 | 3.80 | 3.37 | 2.21 | 2.70 |

Table 2: Quantitative performance of EarthMind on public benchmarks. For image-level evaluation, we report accuracy on AID, UC-Merced, RSVQA-HRBEN, and the VQA task of VRSBench. For region-level evaluation, we report CIDEr scores on DIOR-RSVG and Acc@0.5 on the Visual Grounding task of VRSBench. For pixel-level tasks, mean Intersection over Union (mIoU) is reported.

| Method | Image-level | | | | Region-level | |
|---|---|---|---|---|---|---|
| | AID | UC | RSVQA | VRS-VQA | RSVG | VRS-VG |
| GPT-4o (OpenAI, 2024) | 74.7 | 88.8 | - | - | - | - |
| LLaVA-1.5 (Liu et al., 2023a) | 72.0 | 84.4 | 63.1 | 76.4 | - | 5.1 |
| GeoChat (Kuckreja et al., 2024) | 72.0 | 84.4 | 72.3 | 76.0 | 30.9 | 49.8 |
| EarthGPT (Zhang et al., 2024b) | - | - | 72.0 | - | 232.8 | - |
| EarthMarker (Zhang et al., 2024a) | 78.0 | 86.5 | - | - | 379.3 | - |
| EarthDial (Soni et al., 2024) | 88.8 | 92.4 | 72.5 | - | - | - |
| **EarthMind** | **97.2** | **95.0** | **74.0** | **78.9** | **428.2** | **55.6** |

| Method | Pixel-level | |
|---|---|---|
| | RRSIS-D | RefSegRS |
| LAVT (Yang et al., 2022) | 56.8 | 47.4 |
| RIS-DMMI (Hu et al., 2023) | 60.3 | 52.2 |
| Caris (Liu et al., 2023b) | 62.1 | 42.7 |
| RM-SIN (Liu et al., 2024b) | 64.2 | 42.6 |
| CroBIM (Dong et al., 2024) | 64.5 | 59.8 |
| GeoPixel (Shabbir et al., 2025) | 67.3 | - |
| **EarthMind** | **82.2** | **62.6** |

three granularities: (1) **Image-level**: classification on AID (Xia et al., 2017) and UCMerced (Yang & Newsam, 2010), and VQA on RSVQA-HRBEN (Lobry et al., 2020) and VRSBench (Li et al., 2024b); (2) **Region-level**: captioning and visual grounding on DIOR-RSVG (Zhan et al., 2023) and VRSBench; (3) **Pixel-level**: referring expression segmentation on RefSegRS (Yuan et al., 2024b) and RRSIS-D (Liu et al., 2024b). Additionally, we evaluate multi-sensor understanding on SAR (Wang et al., 2019), BigEarthNet (Sumbul et al., 2019), and SoSAT-LCZ42 (Zhu et al., 2019) datasets.

## 5.3 RESULTS

**EarthMind-Bench.** EarthMind-Bench supports evaluation under three settings: Optical only, SAR only, and Optical–SAR fusion. For multispectral images, baseline models are evaluated using only RGB channels due to their inability to process multi-channel inputs, while EarthMind can leverage the full spectral information. We compare EarthMind with state-of-the-art EO-specific MLLMs and proprietary models such as GPT-4V (OpenAI, 2023) and GPT-4o (OpenAI, 2024). Tab. 1 summarizes the results, from which we highlight three key findings: **1) Fine-grained and open-ended tasks remain challenging for existing MLLMs.** While coarse-grained tasks like scene classification are largely solved, tasks such as object counting and spatial relationship understanding still exhibit significant performance gaps. In particular, referring expression segmentation proves difficult for most models due to the lack of pixel-level reasoning ability. **2) Most models generalize poorly to SAR inputs.** Performance under SAR-only settings lags behind Optical settings for nearly all models, likely due to training data limitations. In contrast, EarthMind demonstrates strong generalization to SAR inputs, benefiting from multi-sensor training. **3) Effective fusion requires learning cross-

Table 3: Quantitative performance on multispectral and SAR benchmarks. Evaluation follows the protocol of (Soni et al., 2024). † indicates that the results on benchmarks were reproduced using their official weights.

| Method | Multispectral | |
|---|---|---|
| | BigEarthNet | SoSAT-LCZ42 |
| GPT-4o (OpenAI, 2024) | 49.0 | 15.5 |
| Qwen2.5-VL†(Bai et al., 2025) | 36.2 | 18.9 |
| EarthDial (Soni et al., 2024) | 69.9 | **60.7** |
| **EarthMind** | **70.4** | 58.3 |

| Method | SAR | | | | |
|---|---|---|---|---|---|
| | Small | Medium | Large | Single | Multiple |
| GPT-4o (OpenAI, 2024) | 0.70 | 0.90 | 3.20 | 1.20 | 0 |
| GeoChat†(Kuckreja et al., 2024) | 2.61 | 4.92 | 6.95 | 2.58 | 1.40 |
| EarthDial (Soni et al., 2024) | 12.14 | 26.02 | 35.56 | 26.03 | 6.06 |
| **EarthMind** | **13.58** | **28.55** | **36.78** | **27.45** | **6.99** |

Table 4: (**Left**) Comparison of our proposed HCA methods with naive fusion methods on EarthMind-Bench, including Scene Classification (SC), Object Existence (OE), Hallucination Detection (HD), Object Counting (OC), Spatial Relationship (SR) and Referring Expression Segmentation (RS). (**Right**) Modality Attention Gap (Optical - SAR) across layers. For concatenation, the gap is computed from MAS using Eq. 1. For HCA, the gap is derived from the learned fusion weights ($w_i^o$ and $w_i^s$) during cross-modal attention. Lower gaps indicate better modality balance.

| Method | SC | OE | HD | OC | SR | RS | Avg |
|---|---|---|---|---|---|---|---|
| Single Modality | 64.4 | 77.5 | 83.6 | 50.1 | 43.1 | 56.0 | 62.5 |
| Concatenation | 64.2 | 77.9 | 83.3 | 49.6 | 34.2 | 57.6 | 61.1 |
| Element-wise Average | 64.7 | 79.3 | 85.4 | 49.8 | 36.9 | 57.0 | 62.2 |
| Native Attention | 65.0 | 79.9 | 85.2 | 49.8 | 38.7 | 58.2 | 62.8 |
| **HCA** | **65.5** | **84.4** | **88.1** | **52.4** | **59.7** | **59.8** | **68.3** |

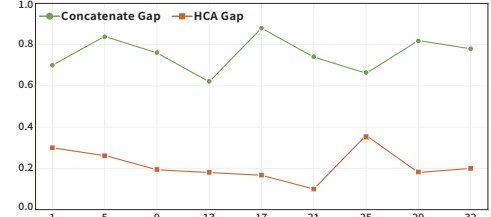

**modal complementarity, not just stacking modalities.** As the only open-source model with explicit fusion capability, EarthMind is compared against GPT-4 variants which feed both Optical and SAR data as pseudo-RGB inputs using the official multi-image interface. Results show that GPT-4 models experience performance degradation compared to Optical-only inputs, particularly on fine-grained tasks like Route Planning, Object Counting, and Spatial Relationship Understanding, where SAR data provides robust structural cues under adverse conditions. In contrast, EarthMind effectively captures cross-modal complementarity, yielding consistent improvements over single-modality inputs.

**Public Benchmarks.** We evaluate EarthMind on mainstream EO benchmarks (Tab. 2). EarthMind consistently delivers strong performance across multi-level understanding tasks. On image-level tasks (AID, UC-Merced, RSVQA-HRBEN, VRSBench-VQA), it significantly outperforms previous models, including GPT-4o, despite using only 4B parameters. For region-level tasks, EarthMind achieves 428.2 CIDEr on DIOR-RSVG and 55.6% accuracy on VRSBench visual grounding. On pixel-level benchmarks, it achieves top results on both RRSIS-D and RefSegRS, surpassing specialized segmentation models. Beyond RGB data, EarthMind demonstrates competitive results on SAR and multispectral benchmarks (Tab. 3), showing strong generalization to diverse EO scenarios.

## 5.4 ABLATIONS

EarthMind significantly enhances multi-sensor EO data understanding through the proposed Hierarchical Cross-modal Attention (HCA) module and the curated FusionEO dataset. We examine each component below, with additional analysis in Appendix H.

**The effectiveness of HCA.** To evaluate the effectiveness of our HCA module, we conduct ablations on the multi-sensor setting by comparing with the better single-modality performance (optical or SAR) and three fusion configurations: (1) *Concatenate*: visual tokens from different modalities are directly concatenated and passed to the LLM; (2) *Element-wise Averaging*: visual tokens are summed and averaged to obtain fused representations; (3) *Naive Attention*: token importance is computed using cosine similarity between paired SAR and optical features. We report both multiple-choice task performance and referring expression segmentation results on EarthMind-Bench.

As shown in Tab. 4, our method significantly outperforms all naive fusion methods, especially on challenging tasks like object counting and spatial relationship reasoning. These fine-grained tasks require detailed visual perception and reasoning, making them more susceptible to poor optical image quality and thus benefiting from complementary information provided by SAR imagery. To understand the underlying mechanism, we calculate the Modality Attention Score (MAS) proposed in

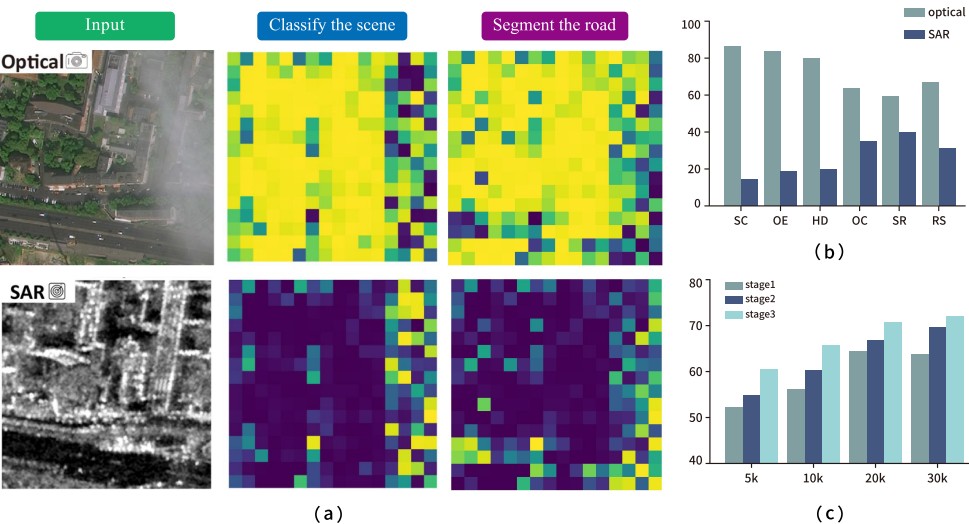

Figure 4: (**a**) Visualization of learned attention weights of optical and SAR under the different queries. (**b**) Statistics of attention weights in different tasks. (**c**) The ablations of the proposed FusionEO data.

Eq. 1 and visualize the results for HCA and naive concatenation across different layers. The analysis reveals a dramatic attention imbalance in naive concatenation: optical tokens receive significantly higher attention than SAR tokens, creating a substantial attention gap throughout all layers. This confirms that the model is nearly blind to SAR information. In contrast, our HCA method significantly reduces this attention gap, achieving a more balanced distribution. This rebalancing is particularly pronounced in shallow layers where fundamental perceptual representations are established, enabling the model to effectively leverage complementary information from both modalities.

We also visualize a typical case in Fig. 4 (a), which reveals interesting findings. First, low-quality regions in optical images receive low attention weights, where SAR can provide complementary information. Moreover, the attention weights are highly relevant to the query: for global tasks like classification, the model primarily relies on the optical modality for scene understanding, while for fine-grained tasks like segmentation, SAR modality gains increased importance, especially in query-relevant regions. For instance, when segmenting roads, the model learns to leverage complementary information from both modalities, with SAR providing crucial structural details that may be obscured or ambiguous in optical imagery. We further calculate the task-specific fusion weights using Eq. 7 and visualize the results in Fig. 4 (b) to elaborate this phenomenon, where optical modality dominates in semantic-rich tasks while SAR modality begins to contribute significantly in fine-grained tasks.

**The effectiveness of FusionEO.** Our proposed FusionEO employs a three-stage curation pipeline. To validate its effectiveness, we report the performance scaling on EarthMind-Bench in Fig. 4 (c). As shown, our curated data achieves consistent performance gains with increasing data scale. Compared to Stage 1 data which contains coarse-grained information, the RoI information injection in subsequent stages provides more fine-grained supervision. Building upon this foundation, our final dataset generates diversified QA pairs, thus achieving the best generalization performance.

# 6 CONCLUSION

In this work, we propose EarthMind, a unified framework for multi-sensor Earth Observation data understanding, specifically designed for optical and SAR data fusion. We introduce the Hierarchical Cross-modal Attention (HCA) module, which adaptively selects task-related cross-sensor representations for enhanced LLM reasoning. EarthMind handles diverse EO tasks from image-level understanding to pixel-level segmentation. We develop FusionEO, a 20K paired instruction-tuning dataset, and curate EarthMind-Bench, the first benchmark for multi-sensor fusion in MLLMs. Extensive experiments demonstrate that EarthMind achieves state-of-the-art performance while consistently outperforming existing MLLMs across diverse EO benchmarks, validating its effectiveness and strong generalization.

ETHICS STATEMENT

We acknowledge and adhere to the ICLR Code of Ethics in all aspects of this research. This work presents several ethical considerations that we address below.

**Dataset Usage and Privacy.** All datasets used in this research are publicly available and have been used in accordance with their original licensing terms and usage agreements. The Earth Observation imagery used does not contain personally identifiable information, as the spatial resolution is insufficient to identify individuals. We have properly cited all data sources and maintained the original train/test splits to ensure fair comparison with existing work.

**Dual-Use Technology Considerations.** While Earth Observation technologies can have beneficial applications in environmental monitoring, disaster response, and agricultural planning, we acknowledge that remote sensing capabilities could potentially be misused for surveillance purposes. However, our work focuses on advancing scientific understanding of multi-sensor data fusion for legitimate environmental and scientific applications. The datasets and benchmarks we develop are intended for the research community to advance Earth system science.

**Environmental Impact.** We acknowledge that training large models requires significant computational resources with associated carbon footprint. To mitigate this impact, we have designed our architecture to be efficient (4B parameters) compared to larger alternatives, and we provide detailed computational requirements to help researchers make informed decisions about resource usage.

**Bias and Fairness.** Our training data is sourced from publicly available Earth Observation datasets that may contain geographical biases toward certain regions. We acknowledge this limitation and encourage future work to ensure more global representation in Earth Observation datasets.

**Open Science.** To promote transparency and reproducibility, we commit to releasing our code, model weights, and detailed experimental protocols upon publication, enabling the research community to verify and build upon our contributions.

REPRODUCIBILITY STATEMENT

To ensure reproducibility, we provide comprehensive details across multiple components of our work. The EarthMind architecture and Hierarchical Cross-modal Attention (HCA) module are fully described in Sec. 3 with complete mathematical formulations. Implementation details, including model configurations, training hyperparameters, and optimization settings, are provided in Appendix F. The FusionEO dataset construction pipeline is detailed in Sec. 3.3 with the three-stage automated curation process fully specified. EarthMind-Bench construction and evaluation protocols are comprehensively documented in Sec. 4 and Appendix E. All public datasets used in our experiments are properly cited with their official splits maintained to ensure fair comparison. Code for model training, evaluation scripts, and data processing pipelines will be made available upon publication to facilitate reproduction of our results. The computational requirements and training infrastructure details are specified in Appendix D to aid in replication efforts.

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

## A  OVERVIEW OF APPENDIX

## B  THE USE OF LARGE LANGUAGE MODELS (LLMS)

In accordance with ICLR policies on LLM usage, we disclose that no Large Language Models were used in any aspect of this research work. All content, including research conception, methodology development, experimental design, code implementation, data analysis, result interpretation, and manuscript preparation, was conducted entirely by the authors without LLM assistance. The authors assume full responsibility for all content, claims, and conclusions in this paper.

## C  DISCUSSIONS AND BROADER IMPACT

In this section, we illustrate the limitations, the distinction between our method and existing ones, as well as the broader impact.

**Limitation and Future Work.** Training EarthMind requires considerable computational resources due to its use of multiple visual encoders for multi-level understanding. A promising direction is to optimize the architecture via Mixture-of-Experts or knowledge distillation to reduce redundancy. Additionally, a modality-aligned encoder that jointly embeds heterogeneous sensor inputs into a shared semantic space could further improve efficiency. Moreover, our EarthMind-Bench currently contains only optical-SAR data; future work will incorporate additional sensing modalities such as multispectral, hyperspectral, and infrared imagery. Furthermore, benchmarking diverse fusion scenarios involving more than two modalities would better reflect real-world EO challenges.

Table A: **(Left)** Comparison of EarthMind with existing EO LMMs. EarthMind supports both multi-granular and multi-sensor understanding. **(Right)** Comparison of EarthMind-Bench with existing EO benchmarks in terms of multi-sensor support, granularity, and task level. "S" denotes single-modality input; "M" indicates multiple modalities (used independently); "F" represents paired sensor fusion. "MC" and "OE" refer to multiple-choice and open-ended formats, respectively.

| Method | Multi-Granular | | | Multi-Sensor | |
|---|---|---|---|---|---|
| | Image | Region | Pixel | Handling | Fusion |
| RSGPT (Hu et al., 2025) | ✓ | ✗ | ✗ | ✗ | ✗ |
| GeoChat (Kuckreja et al., 2024) | ✓ | ✓ | ✗ | ✗ | ✗ |
| EarthGPT (Zhang et al., 2024b) | ✓ | ✓ | ✗ | ✓ | ✗ |
| Earthmarker (Zhang et al., 2024a) | ✓ | ✓ | ✗ | ✗ | ✗ |
| LHRS-bot (Muhtar et al., 2024) | ✓ | ✗ | ✗ | ✗ | ✗ |
| SkyEyeGPT (Zhan et al., 2025) | ✓ | ✗ | ✗ | ✗ | ✗ |
| Skysensegpt (Zhan et al., 2025) | ✓ | ✗ | ✗ | ✗ | ✗ |
| EarthDial (Soni et al., 2024) | ✓ | ✗ | ✗ | ✓ | ✗ |
| GeoPixel (Shabbir et al., 2025) | ✓ | ✗ | ✓ | ✗ | ✗ |
| RSUniVLM (Liu & Lian, 2024) | ✓ | ✓ | ✓ | ✗ | ✗ |
| **EarthMind** | ✓ | ✓ | ✓ | ✓ | ✓ |

| Benchmark | Multi Sensor | Multi Gran. | Multi Level | Task Type |
|---|---|---|---|---|
| RSIEval (Hu et al., 2025) | S | ✗ | ✓ | MC + OE |
| HnstD (Pang et al., 2025) | S | ✗ | ✗ | MC + OE |
| GeoChat-Bench (Kuckreja et al., 2024) | S | ✗ | ✗ | OE |
| VRSBench (Li et al., 2024b) | S | ✗ | ✗ | MC + OE |
| LHRS-Bench (Muhtar et al., 2024) | S | ✗ | ✓ | MC |
| VLEO-Bench (Zhang & Wang, 2024) | S | ✗ | ✓ | MC + OE |
| FIT-RSRC (Luo et al., 2024) | S | ✗ | ✓ | MC |
| UrBench (Zhou et al., 2025) | S | ✓ | ✓ | MC |
| XLRS-Bench (Wang et al., 2025) | S | ✓ | ✓ | MC + OE |
| GEOBench-VLM (Danish et al., 2024) | M | ✓ | ✓ | MC + OE |
| **EarthMind-Bench** | **M + F** | ✓ | ✓ | MC + OE |

**Comparison with Existing Works.** The key distinction of our work from existing models is that EarthMind is the first to address both multi-sensor fusion and multi-granular tasks simultaneously. For multi-sensor capability, our model can flexibly support single or multiple visual inputs; for multi-granular understanding, it can handle multiple levels of granularity, from pixel-level segmentation and region-level semantic understanding to image-level scene classification. As summarized in Tab. A, achieving fine-grained, multi-sensor comprehension in EO remains largely unresolved by existing approaches.

**Broader Impact.** EarthMind offers significant impact to both the large multimodal model (LMM) community and practical applications. Methodologically, it demonstrates how vision-language models can be extended to handle multi-granular tasks via the proposed Hierarchical Cross-modal Attention mechanism, which adaptively allocates attention to task-relevant regions and modalities. This architectural design extends beyond Earth Observation and can be generalized to other domains requiring fine-grained spatial reasoning, such as medical imaging and autonomous driving. For the EO community, EarthMind contributes both algorithmically and empirically: it introduces an effective multi-sensor fusion framework and proposes a scalable training pipeline, accompanied by a comprehensive benchmark and instruction-tuning dataset. These contributions can benefit various downstream applications in remote sensing, including disaster forecasting, infrastructure monitoring, and environmental assessment.

# D    DETAILS OF EARTHMIND

EarthMind is built upon the InternVL-2 framework (Chen et al., 2024b). In InternVL-2, each image is divided into multiple patches at pre-defined scales. Each patch is processed by the visual encoder and encoded into 256 tokens. For instance, an image with 4 patches (plus a global image token) yields $(4 + 1) \times 256$ visual tokens in total. To support multi-sensor input, non-optical imagery (e.g., SAR or multispectral data) is transformed into a synthetic video-like sequence by stacking pre-processed frames. This sequence is then concatenated before the input query, following the protocol of multi-frame vision-language models.

In addition, we extend the tokenizer by introducing a special "[SEG]" token for pixel-level grounding. The hidden state of the final LLM layer corresponding to the "[SEG]" token serves as the semantic prompt for mask generation. During inference, if the "[SEG]" token is not generated, we interpret it as an indication that the queried object is not present in the image.

# E    DETAILS OF EARTHMIND-BENCH

**Overview.** We present a more detailed analysis of EarthMind-Bench in Fig. A. Subfigures (a–c) illustrate the distribution of task categories, while (d) and (e) report the number of samples per task. Notably, our benchmark includes 438 referring expression segmentation samples paired with corresponding binary masks, highlighting its support for fine-grained pixel-level grounding. We further visualize the word clouds of questions (f) and answers (g), showing that EarthMind-Bench

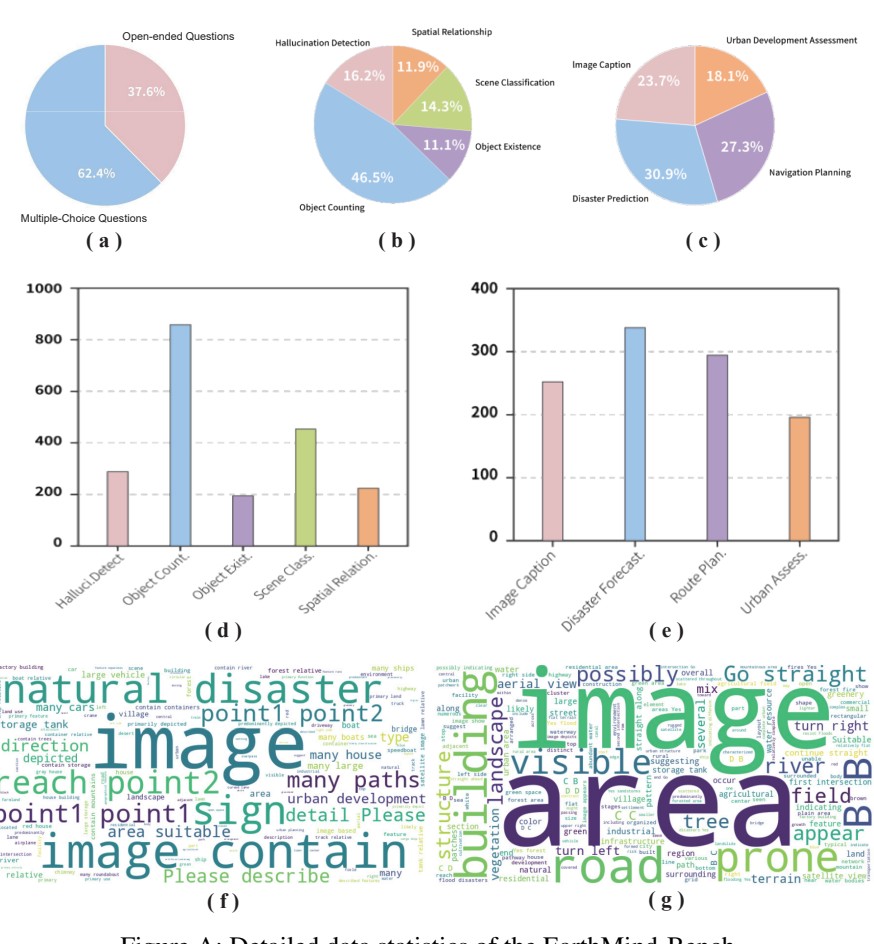

Figure A: Detailed data statistics of the EarthMind-Bench.

covers a wide variety of object types and semantics, enabling comprehensive evaluation across perception and reasoning tasks.

**Data Annotations.** Among the 10 subtasks in EarthMind-Bench, annotations for Scene Classification and Referring Expression Segmentation are directly inherited from their original datasets. For the remaining 8 tasks, we rely on human annotations. We recruit 8 domain experts in geoscience, each assigned to annotate a specific task. After the initial annotation phase, all experts cross-validate each other's work and score the quality of samples. Only high-quality samples with consensus are retained in the final benchmark. To reduce the annotation burden, we first leverage GPT-4o to generate detailed image descriptions, which annotators can reference when constructing question-answer pairs. To ensure consistency and annotation fidelity, we also provide strict task-specific annotation guidelines for all subtasks, which are shown as follows:

- **1. Object Existence.** Determine whether a specific object or class exists in the image (e.g., Is there a bridge in the image?"). *Guideline:* The object should be clearly visible and occupy a non-trivial area. If heavily occluded or ambiguous, mark as Not present."

- **2. Hallucination Detection.** Detect whether the model falsely recognizes objects that do not exist. *Guideline:* Ask for fine-grained distinctions between semantically similar objects (e.g., "Is there a train or a bus in the image?"). Confirm that the mistaken object does not exist even under partial occlusion.

- **3. Object Counting.** Count the number of objects from a given category (e.g., "How many buildings are visible in the image?"). *Guideline:* Only count objects that are visually distinguishable and not clustered into ambiguous shapes. Accept a small margin of error (±1) in complex scenes.

- **4. Spatial Relationship.** Identify relative positions between objects (e.g., What is next to the waterbody?"). *Guideline:* Use cardinal or contextual spatial relations (e.g., to the left

of," adjacent to," surrounded by"). Annotators should verify object co-existence and relative proximity.

- **5. Route Planning.** Generate or select feasible navigation routes from a start to a target location. *Guideline:* Ensure the proposed path avoids obstacles (e.g., rivers, buildings), follows terrain constraints (e.g., avoid steep hills), and adheres to logical movement (e.g., roads preferred over fields).

- **6. Image Captioning.** Generate descriptive sentences summarizing the content and layout of the image. *Guideline:* Captions should mention key objects, land types, spatial layout, and human-made structures if visible. Avoid hallucination and be concise yet informative.

- **7. Disaster Forecasting.** Assess the likelihood of disaster based on visual evidence (e.g., "Is this area prone to flooding?"). *Guideline:* Use cues such as terrain (lowlands), proximity to water, lack of infrastructure, or signs of previous disaster impact. Avoid speculative answers.

- **8. Urban Development Assessment.** Evaluate whether an area is suitable for urban development. *Guideline:* Consider factors such as flat terrain, absence of natural barriers, existing road access, and land cover type. Annotators should justify suitability with visual cues.

**Evaluation Metrics.** For multiple-choice tasks, we evaluate model performance using standard accuracy, measuring the percentage of predictions that exactly match the ground-truth option. For open-ended tasks, inspired by (Zhou et al., 2024), we adopt a GPT-4-based evaluation protocol that assesses the alignment between the model-generated answers and human annotations. As illustrated in Fig. B, we prompt GPT-4 to score the responses based on semantic similarity and correctness, following a structured rubric to ensure consistent evaluation across tasks.

---

**Evaluation Prompt For EarthMind–Bench open–ended Task**

**###Task Description:** You are required to evaluate a respondent's answer based on a provided question, some scoring points,
and the respondent's answer. You should provide two scores. The first is the accuracy score, which should range from 1 to 5. The
second is the relevance score, which should also range from 1 to 5. Below are the criteria for each scoring category.

**###Scoring Criteria:** Please rate the similarity between the **predicted caption** and the
**ground truth** based on the following criteria:1 - Completely unrelated (content is very different)
2 - Slightly related, but most descriptions do not match
3 - Somewhat similar, with a few common details but also clear differences
4 - Mostly matching, only a few minor differences
5 - Highly consistent, both descriptions describe the same content in detail
**##INSTRUCTION:**
Output Scores in JSON Format: Present the scores in JSON format as follows...

Figure B: The illustration of open-ended task evaluation prompt.

# F    DETAILS OF EXPERIMENTAL SETTINGS

We elaborate on the training and inference details of EarthMind. Specifically, we report the hyperparameters in the fine-tuning stage, as shown in Tab. B.

# G    TRAINING DATASET

EarthMind is trained on large-scale natural image datasets, including LLaVA-665K (Liu et al., 2023a), 56K referring expression data (Kazemzadeh et al., 2014; Yu et al., 2016), and 214K grounding

| Hyperparameter | Value |
|---|---|
| Overall batch size | 64 |
| Learning rate | 4e-5 |
| LR Scheduler | Cosine decay |
| DeepSpeed ZeRO Stage | ZeRO-2 |
| Optimizer | Adam |
| Warmup ratio | 0.3 |
| Epoch | 1 |
| Weight decay | 0 |
| Precision | bf16 |

Table B: Hyperparameters of EarthMind.

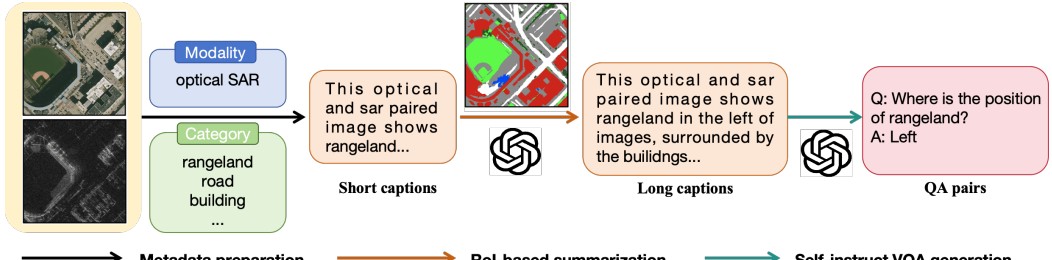

Figure C: The pipeline of FusionEO curation.

conversation generation samples (Rasheed et al., 2024). These datasets cover image-level captioning, VQA, and text-driven segmentation. We also incorporate 724K region-level descriptions from the Osprey dataset (Yuan et al., 2024a) to improve region-level understanding capacity. Second, we introduce EO-specific multimodal data to adapt EarthMind to the remote sensing domain. This includes 1M VQA data from EarthGPT (Zhang et al., 2024b), 142K EO conversations from VRS-Bench (Li et al., 2024b), 31K region-level captions from DIOR-RSVG (Zhan et al., 2023), and 21K referring segmentation samples from RRSIS-D (Liu et al., 2024b) and RefSegRS (Yuan et al., 2024b). Moreover, we involve 500k multi-spectral data, including BigEarthNet (Sumbul et al., 2019) and SoSAT-LCZ42 (Sumbul et al., 2019). Third, we utilize our synthesized multi-sensor conversation corpus (20K RGB-SAR paired dialogues) and selectively retain examples from earlier stages to mitigate catastrophic forgetting.

**Paired Multi-Sensor Training Data Curation.** We construct a high-quality paired optical-SAR training corpus from six publicly available datasets: BigEarthNet-MM (Sumbul et al., 2021), OpenEarthMap-SAR (Xia et al., 2025), DFC2023 Track2 (Persello et al., 2023), WHU-OPT-SAR (Li et al., 2022), MSAW (Shermeyer et al., 2020), and MultiResSAR (Zhang et al., 2025b). All optical-SAR pairs are sampled from their training splits to avoid overlap with EarthMind-Bench.

To scale data generation, we design an automatic three-stage synthetic pipeline leveraging GPT-4o, as shown in Fig. C: **Stage 1: Metadata Preparation** involves extracting modality information (optical and SAR) and category labels (e.g., rangeland, road, building) for each image pair. **Stage 2: RoI-based Summarization** uses the optical image and metadata as input to GPT-4o to generate comprehensive scene descriptions. The model produces both short captions summarizing key objects and spatial relationships, and detailed long captions that provide fine-grained descriptions of regions of interest and their contextual relationships. **Stage 3: Self-instruct VQA Generation** creates diverse question-answer pairs based on the generated captions and multi-sensor context. These cover various reasoning types including: (1) Object Existence (e.g., "Is there a river in the image?"), (2) Counting (e.g., "How many buildings are visible?"), (3) Spatial Relations (e.g., "What is next to the forest?"), (4) Object Localization (e.g., "Where is the road located?"), and (5) Scene-Level Understanding (e.g., "Is this area suitable for agriculture?").

All outputs are structured in JSON format for seamless integration. This three-stage approach ensures that the generated instruction data captures both coarse-grained scene understanding and fine-grained

spatial reasoning across modalities. In total, we curate 30K synthetic multi-sensor samples as our FusionEO dataset for training cross-sensor MLLMs.

Table C: Ablations on the non-rgb data processing.

| Method | Multispectral | | | Method | SAR | | | | |
|---|---|---|---|---|---|---|---|---|---|
| | BigEarthNet | SoSAT-LCZ42 | | | Small | Medium | Large | Single | Multiple |
| Three-band grouping | 70.4 | 58.3 | | Zero-padding | 12.14 | 26.02 | 35.56 | 26.03 | 6.06 |
| Single-band grouping | 71.2 | 59.2 | | Channel replication | 11.08 | 25.99 | 34.38 | 25.99 | 4.32 |

Table D: Ablation study on Multi-granular joint training.

| Method | Image-level (Accuracy) | | | | Region-level | | Pixel-level (mIoU) | |
|---|---|---|---|---|---|---|---|---|
| | AID | UC | RSVQA | VRS-VQA | RSVG | VRS-VG | RRSIS-D | RefSegRS |
| Only trained on Image data | 97.0 | 95.1 | 73.8 | 77.8 | - | - | - | - |
| Only trained on Region data | - | - | - | - | 379.6 | 49.8 | - | - |
| Only trained on segmentation data | - | - | - | - | - | - | 77.6 | 59.3 |
| **EarthMind** | **97.2** | **95.0** | **74.0** | **78.9** | **428.2** | **55.6** | **82.2** | **62.6** |

Table E: Ablation study on joint multi-sensor training.

| Method | EarthMind-Bench | | |
|---|---|---|---|
| | RGB | SAR | RGB+SAR |
| Only trained on RGB | 68.4 | 30.1 | 28.4 |
| Only trained on SAR | 45.6 | 59.8 | 22.3 |
| **trained on paired RGB-SAR** | **69.0** | **67.5** | **70.6** |

# H  MORE ABLATION STUDIES

**Ablation on Multi-Sensor Data Processing.** One of the key strengths of EarthMind lies in its ability to handle multi-sensor data beyond standard RGB imagery. To better understand the impact of different preprocessing strategies, we conduct a series of experiments focused on the handling of SAR and multispectral (MS) data. For SAR inputs with fewer than three channels, we compare two strategies: (1) *Zero padding*, where the missing channels are filled with zeros; (2) *Channel replication*, where existing channels are duplicated to reach three channels. For multispectral data with more than three bands, we evaluate: (1) *Three-band grouping*, where every three consecutive bands are grouped to form one RGB-like frame; (2) *Single-band grouping*, where each band is treated as an individual frame, forming a multi-frame sequence. We evaluate each method under the same training settings and report classification accuracy on the test sets of BigEarthNet (for MS) and SAR Ship Detection (for SAR), as summarized in Tab. C. Our results show that zero padding outperforms channel replication for SAR data, likely because copying channels introduces redundancy and potential noise. For MS data, single-band grouping slightly improves performance due to better spectral resolution, but incurs substantial computational overhead due to the increased number of tokens. Considering both effectiveness and efficiency, we adopt zero padding for SAR and three-band grouping for MS as our default configuration. In the future, we will consider involve token reduction techniques (Liu et al., 2025) to reduce overhead cost.

**Ablation on Joint Multi-Granular and Multi-Sensor Training.** EarthMind is designed to be jointly trained on both multi-granular and multi-sensor data. To validate the effectiveness of this unified training paradigm, we conduct ablation studies along two axes: (1) Multi-Granular Training. We compare two settings: (i) Joint training with image-level, region-level, and pixel-level data. (ii) Independent training for each granularity using the same total amount of data. (2) Multi-Sensor Training. We also compare: (i) Joint training with all available sensor modalities (e.g., RGB, SAR, MS). (ii) Independent training with each modality separately. As shown in Tab. D and Tab. E, the multi-granular co-training strategy consistently outperforms independently trained counterparts, especially on pixel-level tasks. This suggests that high-level semantic supervision (e.g., image-level QA) can improve fine-grained understanding through shared representation learning. Similarly,

Table F: Ablation study on the first stage of curriculum training.

| Method | Image-level (Accuracy) | | | | Region-level | | Pixel-level (mIoU) | |
| --- | --- | --- | --- | --- | --- | --- | --- | --- |
| | AID | UC | RSVQA | VRS-VQA | RSVG | VRS-VG | RRSIS-D | RefSegRS |
| w/o pretraining | 96.5 | 94.8 | 73.2 | 77.6 | 406.7 | 49.8 | 75.4 | 55.3 |
| **with pretraining** | **97.2** | **95.0** | **74.0** | **78.9** | **428.2** | **55.6** | **82.2** | **62.6** |

Table G: Ablation study on the second stage of curriculum learning.

| Method | EarthMind-Bench | | |
| --- | --- | --- | --- |
| | RGB | SAR | RGB+SAR |
| w/o pretraining on RGB EO data | 67.5 | 64.3 | 68.9 |
| **with pretraining on RGB EO data** | **69.0** | **67.5** | **70.6** |

multi-sensor co-training improves generalization across modalities. Notably, the performance under SAR-only evaluation is significantly enhanced by leveraging complementary information learned from RGB and MS data during joint training. This highlights EarthMind's ability to exploit cross-modal synergy in both perception and reasoning tasks.

**Ablation on Curriculum Training Strategy.** EarthMind adopts a curriculum learning strategy that gradually adapts the model from general vision-language data to remote sensing tasks. Specifically, the training proceeds in three stages: (1) pretraining on large-scale natural image VQA and captioning datasets, (2) domain adaptation using EO-specific RGB data, and (3) fine-tuning on multi-sensor data (e.g., SAR). To evaluate the effectiveness of this curriculum, we conduct ablation studies by removing or reordering training stages. As shown in Tab. F and Tab. G, pretraining on general image-language data provides strong foundational capabilities, leading to significant performance improvements on EO tasks, particularly in pixel-level segmentation. Furthermore, RGB-domain training not only enhances performance on RGB inputs but also boosts multi-sensor fusion results, demonstrating its role as an effective bridge between general vision and SAR-specific domains.

# I  MORE VISUALIZATION RESULTS

More visualization results of EarthMind can be seen in Fig. D.

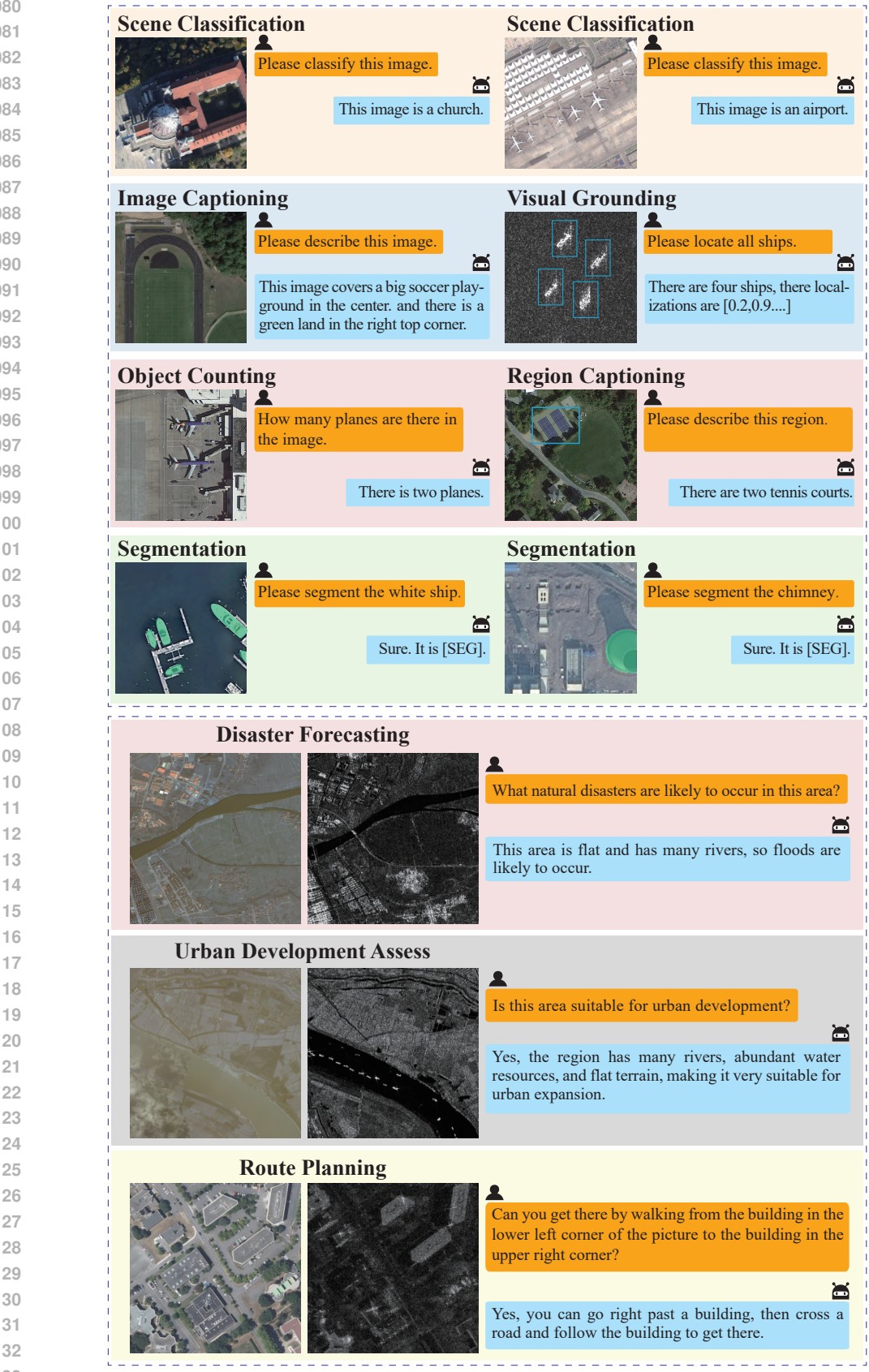

Figure D: More visualization of EarthMind.

