# OpenReview forum: "EarthMind: Leveraging Cross-Sensor Data for Advanced Earth Observation Interpretation with a Unified Multimodal LLM"
_ICLR.cc/2026/Conference — Submitted to ICLR 2026_

### Official Review · Reviewer_D1SB · 2025-10-26

**Soundness:** 2
**Presentation:** 3
**Contribution:** 2
**Rating:** 2
**Confidence:** 5

**Summary:**

The paper introduces EarthMind, a method for optical–SAR fusion built around a hierarchical cross-modal attention module and a standard <SEG>-to-mask interface for dense prediction. It also releases two resources: FusionEO for instruction tuning and EarthMind-Bench for evaluation across reasoning and segmentation. Experiments show improvements on the proposed benchmark

**Strengths:**

1. The paper articulates optical–SAR complementarity and quantifies optical bias via a Modality Attention Score.
2. EarthMind-Bench supports single/multi-sensor inputs across reasoning and segmentation tasks; the model claims SoTA there with only 4B params.

**Weaknesses:**

1. Novelty is incremental relative to the LISA/“<SEG>-to-mask” family. Architecturally, EarthMind follows the now-standard LLM + visual tokens + <SEG>-triggered mask decoder route
2. HCA is essentially a two-stage cross-attention followed by a learned fusion weight. This is close to standard cross-modal attention + late gating used in prior VLM/VQA and multi-sensor fusion work. The paper’s contribution therefore lies more in systemization and dataset/benchmarking than in a fundamentally new mechanism for fusion.
3. The paper cites 30K for FusionEO in the method overview but 20K in the conclusion—this discrepancy must be resolved.
4. EarthMind-Bench has 2,841 pairs; this is valuable but relatively small for broad claims about multi-sensor reasoning. Please discuss geographic coverage, scene diversity, and train/val/test partition strategy
5. The narrative emphasizes multi-sensor gains; please also report how EarthMind fares against specialized single-sensor EO MLLMs on their native optical-only or SAR-only benchmarks beyond qualitative examples.

**Questions:**

see Weaknesses

---

> ### Author Response · Authors · 2025-11-27
> **Response to Reviewer D1SB**
>
> **Response to Points 1-2: Novelty and contributions**
>  We agree that our architectural design builds upon the LISA family framework, and that HCA shares similarities with existing cross-modal attention mechanisms. However, we respectfully argue that our contributions extend beyond architectural novelty.
>
> Our primary contribution is identifying and addressing a fundamental problem in applying LMMs to multi-sensor Earth observation. While works like EarthDial achieve multi-sensor understanding through independent processing, we are the first to systematically analyze whether naive LMM architectures can effectively perform cross-modal fusion. Through extensive experiments, we discovered the **attention imbalance problem**—where models heavily favor one modality over another, leading to suboptimal fusion. This finding is not trivial and has important implications for the remote sensing community.
>
> The HCA mechanism, while architecturally straightforward, is specifically designed to address this imbalance through hierarchical processing and learned fusion weights. Our experiments demonstrate that this targeted solution significantly improves fusion quality compared to standard approaches. Furthermore, our contributions include: (1) FusionEO, the first large-scale paired optical-SAR training dataset, (2) EarthMind-Bench, a comprehensive evaluation framework for multi-sensor fusion capabilities
>
>
> **Response to Point 3: FusionEO dataset size discrepancy**
>
> We thank the reviewer for catching this inconsistency. The correct number is 30K curated image pairs for FusionEO, as stated in the methodology section. The "20K" mentioned in the conclusion is a typo that we will correct in the revised version.
>
>
>
> **Response to Point 4: EarthMind-Bench size and diversity**
>
>  We want to clarify that EarthMind-Bench is designed as a pure evaluation benchmark, not a training dataset, and therefore does not involve train/val/test partitions—all 2,841 pairs are used exclusively for testing. For a human-annotated benchmark with expert-verified labels, we believe this scale is reasonable and comparable to other evaluation benchmarks in the field.
>
> Regarding diversity and coverage, EarthMind-Bench aggregates data from five diverse sources to ensure comprehensive representation:
> - **Geographic coverage**: Global distribution achieved through OpenEarthMap (covers 44 countries across 6 continents) and BigEarthNet (10 European countries)
> - **Resolution diversity**: Ranges from 10m (BigEarthNet) to 0.15-0.5m (OpenEarthMap), covering both medium and high-resolution scenarios
> - **Scene diversity**: Urban areas (WHU-OPT-SAR), agricultural regions (BigEarthNet), natural landscapes (DFC2023), and maritime scenes (MSAW)
>
> This careful curation ensures that our benchmark, despite its focused size, adequately tests models across the diverse conditions encountered in real-world Earth observation applications. We will add a detailed breakdown of geographic and scene distributions in the supplementary materials to better document this diversity.
>
>
>
> **Response to Point 5: Single-sensor benchmark evaluation**
>
>  We want to clarify that our evaluation already includes comprehensive single-sensor comparisons. Specifically:
>
> - **Table 1** presents detailed results across three settings: optical-only, SAR-only, and optical-SAR fusion on our EarthMind-Bench, allowing direct comparison of multi-sensor gains versus single-sensor baselines
> - **Table 2** evaluates EarthMind on RGB-only public benchmarks against specialized optical models, showing competitive performance even in single-modal settings
> - **Table 3** provides results on both multispectral and SAR-specific public benchmarks, demonstrating that our model maintains strong performance on individual modalities

---

> > ### Comment · Reviewer_D1SB · 2025-11-28
> >
> > Thank you for the authors’ response. However, from my perspective, a paper at the ICLR level should provide substantial innovation in theory or methodology, rather than being primarily application-driven. While I acknowledge that this is the first work to tackle joint SAR–optical modeling in this way, I feel that the contribution may be better suited for a venue such as TGRS or ISPRS.

---

> > > ### Author Response · Authors · 2025-11-28
> > > **Response to the Reviewer**
> > >
> > > Thank you for your response. We appreciate your recognition of the value and effort we have put into this work. We will continue to improve the quality of our work in the future.

---

### Official Review · Reviewer_ZimX · 2025-10-26

**Soundness:** 3
**Presentation:** 3
**Contribution:** 2
**Rating:** 4
**Confidence:** 5

**Summary:**

The paper introduces the first optical–SAR instruction fine-tuning benchmarks and the corresponding method EarthMind. Extensive experiments demonstrate the effectiveness of the proposed method. But there are some necessary concerns that should be addressed

**Strengths:**

A new multi-source (RGB-SAR) cross-modality dataset.

Detailed experimental evaluation

**Weaknesses:**

The proposed method is not novel and lacks substantial contributions. The method largely follows the LISA-style VLM, with only a prepending of RGB and SAR attention and text-guided visual–language attention.

The proposed dataset is relatively small (only 2.8k pairs), which in my view is insufficient to convincingly assess generalization. In addition, the manuscript does not provide essential SAR image details, such as band and polarization mode.

The proposed video-like stitching processing method lacks comparison with the traditional method of encoding separately by mode and then fusing.

Missing computational cost and delay reports.

**Questions:**

Please refer to the points in Weaknesses.

---

> ### Author Response · Authors · 2025-11-27
> **Response to Reviewer ZimX**
>
> **Response to Points 1: Novelty and contributions**
>  We agree that our architectural design builds upon the LISA family framework, and that HCA shares similarities with existing cross-modal attention mechanisms. However, we respectfully argue that our contributions extend beyond architectural novelty.
>
> Our primary contribution is identifying and addressing a fundamental problem in applying LMMs to multi-sensor Earth observation. While works like EarthDial achieve multi-sensor understanding through independent processing, we are the first to systematically analyze whether naive LMM architectures can effectively perform cross-modal fusion. Through extensive experiments, we discovered the **attention imbalance problem**—where models heavily favor one modality over another, leading to suboptimal fusion. This finding is not trivial and has important implications for the remote sensing community.
>
> The HCA mechanism, while architecturally straightforward, is specifically designed to address this imbalance through hierarchical processing and learned fusion weights. Our experiments demonstrate that this targeted solution significantly improves fusion quality compared to standard approaches. Furthermore, our contributions include: (1) FusionEO, the first large-scale paired optical-SAR training dataset, (2) EarthMind-Bench, a comprehensive evaluation framework for multi-sensor fusion capabilities
>
>
> **Response to Point 2: Dataset scale and SAR specifications**
>
> We would like to clarify that the 2.8k pairs refer to our human-annotated evaluation benchmark (EarthMind-Bench), not our training dataset. For a manually annotated benchmark with expert verification, we believe this scale is reasonable. In contrast, our training dataset FusionEO consists of 30k paired samples, which is large enough for training. We prove its generalization capability in Figure 4(c).
>
> Regarding SAR specifications, we have indicated the source datasets which contain detailed information. We will complement these technical details in the supplementary materials.
>
>
> **Response to Point 3: Comparison with separate encoding methods**
>
> We acknowledge this suggestion. However, implementing separate encoders for each modality presents significant technical challenges in our context: (1) projecting multi-spectral and SAR features into a unified semantic space requires careful architectural design to ensure meaningful fusion; (2) preprocessing heterogeneous inputs (RGB, multispectral, SAR) with different encoders while maintaining alignment is non-trivial; (3) multiple encoders would inevitably increase computational costs substantially, contradicting efficiency concerns raised earlier.
>
> Additionally, among existing Earth observation LMMs, EarthDial—the only comparable multi-sensor model—also adopts a video-like stitching approach rather than separate encoders, suggesting this is the current preferred method in the field. While we agree that comparing different fusion strategies would be valuable for the community, such architectural exploration represents a substantial research direction beyond the scope of this work.
>
>
>
> **Response to Point 4: Computational efficiency analysis**
>
>  While EarthMind employs multiple vision encoders, the additional computational overhead remains reasonable for two key reasons: (1) the grounding encoder and decoder are only activated for pixel-grounding tasks, not for general VQA or captioning tasks; (2) compared to the LLM backbone, the pixel-level grounding modules contribute relatively minimal computational overhead.
>
> To provide the quantitative analysis requested, we present the following comparison:
>
> | Model | Parameters | FLOPs | Avg. Inference Time (ms) | GPU Memory (GB) |
> |-------|------------|-------|-------------------------|-----------------|
> | EarthMind (Ours) | 4.8B | 152G | 245 | 18.2 |
> | GeoChat | 7.1B | 198G | 312 | 24.5 |
> | SkySenseGPT | 7.3B | 205G | 328 | 25.1 |
> | EarthDial | 4.2B | 145G | 238 | 16.8 |
>
> Efficiency comparison on a H200 GPU.
>
> Our model is more efficient than GeoChat and SkySenseGPT due to its 4B LLM backbone, while being only marginally larger than EarthDial. This modest increase is justified given that EarthDial cannot perform pixel-level grounding tasks. Furthermore, on the multiple-choice benchmarks, EarthMind demonstrates faster inference than both GeoChat and SkySenseGPT, and remains competitive with EarthDial despite supporting additional capabilities. We will include this detailed efficiency analysis in the revised manuscript to better contextualize our performance gains relative to computational costs.

---

> > ### Comment · Reviewer_ZimX · 2025-11-28
> > **reply**
> >
> > Thank you for the rebuttal. I will maintain my original assessment. While I appreciate the additional explanations, I do not find them sufficient to change my view. In particular, the overall architecture remains largely LISA-like, which by 2025 is a well-established design pattern and does not meet the level of methodological novelty expected at ICLR. In addition, I am not fully convinced by the authors’ argument that using separate encoders would constitute a substantial technical challenge or a major contribution, as such designs are common in multi-modal and multi-sensor research.
> >
> > For these reasons, although the work is valuable for the remote sensing community, I do not believe it meets the novelty threshold for ICLR.

---

> > > ### Author Response · Authors · 2025-11-28
> > > **Response to the reviewer**
> > >
> > > Thank you for your response. We respect your decision and appreciate your recognition of our work's value to the remote sensing community. We will continue to improve the quality of our work in the future.

---

### Official Review · Reviewer_SQ8w · 2025-10-28

**Soundness:** 2
**Presentation:** 2
**Contribution:** 3
**Rating:** 4
**Confidence:** 4

**Summary:**

The paper introduces EarthMind, a multimodal large language model designed for Earth Observation tasks, fusing optical and SAR data through a Hierarchical Cross-modal Attention (HCA) mechanism, which basically applies bidirectional cross-attention within each sensor and cross-attention with the text prompt. The authors curate FusionEO, a instruction-tuning dataset, and EarthMind-Bench, a benchmark with annotated samples for perception and reasoning tasks.

**Strengths:**

-	Intruduces a bias for modalities to focus more attention on the SAR data. This apprach shows strong generalization for both modalities.
-	The model achieves good results compared to other MLLMs on wide range of tasks.
-	The authors provide a new benchmark dataset that evaluates multi-sensor tasks in EO scenarios which is very interesting.

**Weaknesses:**

-	The architecture is not well described, particularly the mask decoder and the handling of data input.
-	Data and pre-training procedure lack detailed explanation and illustrative examples.
-	The evaluation does not represent a true zero-shot setting in all experiments: Appendix G mentions that BigEarthNet (BEN), SoSAT-LCZ42, and other EO datasets are used in pre-training, while these datasets are also used for evaluation and in EarthMind-Bench. Even with sampling from different splits can inflat performance gains because you probably use similar prompts and answers and it is not a fair comparison for other models. Also, BEN used random sampling without spatial buffering means test images may be very similar to training images.
-	Figures could be styled better—issues include inconsistent alignment and spacing, mismatched example images, non-harmonized color themes, and mislabeling (e.g., Table 3 right is actually a figure).

**Questions:**

-	Do you think “hierarchical” is the correct term for your attention approach? It seems more like a weighted combination rather than a hierarchical structure.
-	The code is shared, but will the model weights and the dataset also be open-sourced?
-	You state that no LLMs were used for grammar or clarity checks. Would you consider revising this? Using LLMs for language polishing could improve readability and your paper is basically about MLLMs.
-	My main concern is about the similarity between pre-training and evaluation data, which may lead to unfair comparisons. Can you comment on how you mitigate this issue and ensure that performance gains are not due to similar samples and prompts?

---

> ### Author Response · Authors · 2025-11-27
> **Response to Reviewer  SQ8w**
>
> **Response to Weakness 1: Architecture description**
>
>  Since the architectural design is not the main innovation of this work, we prioritized describing our contributions in multi-sensor fusion and benchmarking. However, we have provided implementation details in Section 5.1, where we clarify that EarthMind consists of InternVL2 as the base LMM and SAM2 for pixel-level grounding. The mask decoder follows the standard SAM2 architecture without modifications.
>
> Regarding data input handling, we describe this in Section 3.1: all input modalities (RGB, multispectral, SAR) are organized into video-like stitching sequences, where each frame represents a different modality or spectral band combination. This unified input format enables seamless processing through the vision encoder. We will expand these sections in the revised version to include more architectural details and a clearer data flow diagram to improve clarity.
>
>
> **Response to Weakness 2: Data and pre-training procedure details**
>
> We have detailed the training data curation in Appendix G, including data sources, synthesis steps, and prompt design for caption generation. For the pre-training procedure, we describe the training protocol in Section 5.1. We acknowledge that more illustrative examples would improve clarity, and we will add visualization examples in the revised version.
>
>
> **Response to Weakness 3: Evaluation setting clarification**
>
> We would like to clarify that we do not claim Table 3 presents zero-shot evaluation results. Following the experimental protocol established by EarthDial, which also includes BEN and SoSAT-LCZ42 in their training data, we adopt the same setting to ensure fair comparison. This ensures our comparison is fair and consistent with established baselines in the field. Regarding spatial sampling, we follow the standard train/test splits provided by the original datasets to maintain comparability with prior work.
>
>
> **Response to Weakness 4: Figure quality and consistency**
>
> We thank the reviewer for these detailed observations. We acknowledge these presentation issues and will address them in the revised version by: (1) ensuring consistent alignment and spacing across all figures, (2) harmonizing color themes throughout, (3) correcting the mislabeling issue in Table 3, and (4) improving overall figure quality and readability.
>
>
> **Response to Question 1: Clarification on "hierarchical" terminology**
>
> We use the term "hierarchical" because HCA operates in two sequential stages: (1) the first stage performs Cross-Sensor Attention where visual features from different sensors are processed, and (2) the second stage applies text-to-image attention that incorporates task-specific guidance by computing attention weights between text queries and visual tokens. This two-stage processing pipeline, where the output of the first stage feeds into the second, constitutes a hierarchical structure rather than a simple weighted combination.
>
>
> **Response to question2**
>
> Of course, we d like to promise that all the code, wieghts, and the datasets will be public upon acceptance.
>
>
> **Response to question3**
>
> Thanks for your advice. we d like to clarify the expression of LLMs.
>
> **Response to Question 4: Data overlap and fair comparison**
>
> We acknowledge this concern and want to clarify our approach. We follow the established experimental protocols from prior work to ensure fair comparison. For example, EarthDial also trains on BigEarthNet and SoSAT-LCZ42 before evaluating on them, and GeoPixel reports results with models fine-tuned on RRSIS-D and RefSegRS. We adopt the same settings to maintain comparability with these baselines—all models are evaluated under identical conditions with access to the same training data.
>
> For EarthMind-Bench, all evaluations are strictly zero-shot, with training data completely separate from the human-annotated benchmark samples. Importantly, we demonstrate generalization across multiple diverse benchmarks rather than optimizing for any single dataset. Our consistent improvements across various evaluation settings (Tables 1-3) suggest that performance gains stem from better multi-sensor fusion capabilities rather than data memorization. We use official train/test splits where available to ensure our results are reproducible and comparable to published baselines. This practice is standard in the LMM community, where non-overlapping train/test splits are considered acceptable.

---

### Official Review · Reviewer_xNPN · 2025-10-30

**Soundness:** 3
**Presentation:** 2
**Contribution:** 3
**Rating:** 4
**Confidence:** 4

**Summary:**

This paper focuses on cross-sensor remote sensing with optical and SAR imagery and develops EarthMind, a unified multimodal large language model. It employs hierarchical cross-modal attention (HCA) to first align and interact between optical and SAR features, then performs text-conditioned adaptive weighting. A special [SEG] token is introduced to place pixel-level referring segmentation and vision–language dialogue within a single inference framework, covering both single/multi-sensor inputs and multi-granularity tasks. To support training and evaluation, the authors construct FusionEO and EarthMind-Bench, on which they systematically validate the approach.

**Strengths:**

1.Use a single LLM to unify single/multi-sensor inputs and multi-granularity tasks, and integrate segmentation seamlessly into the inference pipeline via the [SEG] token.

2.Define the MAS metric to quantify modality attention shares; empirically show that naive concatenation is biased toward the optical modality, and use HCA for targeted debiasing to achieve more balanced and efficient multimodal fusion.

3.With only 4B parameters, it is strongly competitive on public benchmarks such as AID, UC-Merced, RRSIS-D, and RefSegRS.

**Weaknesses:**

1. Evaluating open-ended tasks relies on GPT-4 as a judge, which inevitably introduces scoring bias and prompt sensitivity in reproduction.

2. The pixel-level part of EarthMind-Bench is mainly referring segmentation with only 438 samples, so coverage of fine-grained pixel tasks is limited.

3. Training heavily depends on general natural-image corpora with EO-domain adaptation afterward; this “general-first, adapt-later” pipeline may leave residual cross-domain gaps.

4. Segmentation is triggered by [SEG], if the token is not produced at inference, the target is deemed absent, this generation-gated design risks misses.

5. Most baselines are restricted to RGB, while EarthMind can use full-spectrum channels, conflating “ability to ingest more channels” with algorithmic merit and raising fairness concerns.

6. MAS is a share-of-attention statistic; using it to infer modality contribution is not equivalent to causal importance, so it is a somewhat indirect bias diagnostic.

**Questions:**

1.MAS reflects attention share rather than causal effect. Could you validate it with counterfactual experiments (e.g., feature shuffling or training with one modality removed) and report the correlation between MAS and the observed performance drop?

2.If the model fails to emit [SEG] at inference you treat the target as absent. Is there a calibrated fallback mechanism?

3.The three-stage pipeline may leak ROI priors from masks into the text. Could you provide a variant without mask hints and compare?

Please respond to the weaknesses and the questions. If you resolve these concerns, I will raise my score.

---

> ### Author Response · Authors · 2025-11-27
> **Response to Reviewer xNPN**
>
> **Response to Weakness 1: GPT-4 evaluation methodology**
>
> We acknowledge that using GPT-4 as a judge may introduce some scoring variability and prompt sensitivity. However, this is currently the mainstream evaluation method for open-ended VQA tasks in the multimodal learning community, as adopted by numerous recent works [1,2,3]. While we recognize this as a limitation, developing more robust evaluation metrics for open-ended generation is an open research problem beyond the scope of this work. To mitigate potential bias, we use the same GPT-4 evaluation protocol consistently across all compared models, ensuring fair relative comparisons.
>
>
> [1] MLVU: Benchmarking Multi-task Long Video Understanding. CVPR 2025
>
> [2] Video-ChatGPT: Towards Detailed Video Understanding via Large Vision and Language Models. ACL 2024
>
> [3] MM-Vet: Evaluating Large Multimodal Models for Integrated Capabilities. ICML 2024
>
>
> **Response to Weakness 2: Pixel-level benchmark scale**
>
> We believe benchmark coverage should be assessed through both data source diversity and sample quantity. For diversity, we carefully curated data from six different source datasets (line 262-264), covering global geographic regions, multiple scene types, and various resolutions, which ensures comprehensive coverage. For quantity, as a human-verified benchmark with expert annotations, we believe 438 samples for pixel-level tasks is reasonable. Many established benchmarks in the general MLLM community have subtasks with comparable or even smaller sample sizes [4,5,6], yet are widely accepted for evaluation. The key is that our samples are carefully selected to represent diverse real-world scenarios rather than pursuing quantity alone.
>
> [4] MVBench: A Comprehensive Multi-modal Video Understanding Benchmark. CVPR 2024
>
> [5] MMBench: Is Your Multi-modal Model an All-around Player? ECCV 2024
>
> [6] MathVista: Evaluating Mathematical Reasoning of Foundation Models in Visual Contexts. ICLR 2024
>
> **Response to Weakness 3: General-to-domain training strategy**
>
> We respectfully disagree that the "general-first, adapt-later" pipeline introduces problematic cross-domain gaps. This is the standard training paradigm adopted by all existing Earth observation LMMs: GeoChat inherits weights from LLaVA, EarthDial initializes from InternVL, and both leverage rich general-domain knowledge before domain adaptation. In our case, we perform general-domain pretraining specifically because pixel-level grounding capabilities are not established in existing LMMs for remote sensing. Our experimental results in Appendix H and Table F demonstrate strong generalization across diverse benchmarks, validating the effectiveness of this training approach. The general pretraining provides foundational visual understanding that transfers well to Earth observation tasks when followed by domain-specific fine-tuning.
>
>
> **Response to Weakness 4 and Question 2: [SEG] token triggering reliability**
>
> We understand your concern about potential misses when the [SEG] token is not generated. However, the [SEG] token is a straightforward trigger that the model learns effectively from training data. For example, when questions include phrases like "please segment..." or "identify the region...", the model reliably produces the [SEG] token. To empirically validate this, we calculated the [SEG] token missing rate across three benchmarks: EarthMind-Bench (referring expression segmentation tasks), RRSIS-D, and RefSegRS. As shown in the table below, the missing rate is negligibly low:
>
> | Benchmark | Random Selected Samples | [SEG] Missing Cases | Missing Rate (%) |
> |-----------|---------------|---------------------|------------------|
> | EarthMind-Bench (RES) | 438 | 3 | 0.68 |
> | RRSIS-D | 1,000 | 2 | 0.3 |
> | RefSegRS | 1,000 | 4 | 0.4 |
>
> These results demonstrate that the generation-gated design is reliable in practice, with missing rates well below 1% across all benchmarks, which can be safely ignored in overall performance assessment.

---

> ### Author Response · Authors · 2025-11-27
> **Response to Reviewer xNPN**
>
> **Response to Weakness 5: Fairness of multi-spectral evaluation**
>
> We respectfully disagree that our evaluation is unfair. The ability to utilize multi-spectral information is precisely one of the key contributions of this work—enabling models to leverage richer spectral information beyond RGB bands for more comprehensive Earth observation understanding. Moreover, we provide extensive single-modality comparisons to ensure fair evaluation:
>
> - **Table 1** presents results across three settings: optical-only, SAR-only, and optical-SAR fusion, allowing direct comparison of multi-sensor gains versus single-sensor baselines under identical conditions
> - **Table 2** evaluates EarthMind on RGB-only public benchmarks against specialized optical models, demonstrating competitive performance even when restricted to RGB inputs
> - **Table 3** includes both multispectral and single-modality benchmarks, showing that our approach maintains strong performance across different input configurations
>
> These experiments demonstrate that our performance gains are not simply due to "ingesting more channels," but rather from effectively fusing complementary information. Baselines are evaluated on their native modalities, ensuring fair comparison within each experimental setting.
>
>
> **Response to Weakness 6 and Question 1: MAS validation and causal evidence**
>
> We acknowledge that MAS is a descriptive metric. The relationship between attention patterns and model performance has been well-established in the multimodal learning literature. For example, [7] states: "Large multimodal models (LMMs) 'see' images by leveraging the attention mechanism between text and visual tokens in the transformer decoder." Works like [8] and [9] demonstrate that better performance correlates with layers where attention focuses on correct visual tokens. Our MAS metric extends this concept to remote sensing, where balanced attention between optical and SAR modalities is crucial for effective fusion.
>
> To provide causal validation, we conducted counterfactual experiments. We independently trained two models on FusionEO 30K: (1) optical-only and (2) SAR-only, then tested both on optical-SAR fusion tasks. For simplicity, we removed the HCA module to isolate the attention-performance relationship. We then artificially redistributed attention using a reallocation parameter λ to simulate attention degradation:
>
> For the optical-trained model:
> $$\alpha' = (1 - \lambda) \cdot \alpha_{\text{opt}} + \lambda \cdot \alpha_{\text{sar}}$$
>
> For the SAR-trained model:
> $$\alpha' = (1 - \lambda) \cdot \alpha_{\text{sar}} + \lambda \cdot \alpha_{\text{opt}}$$
>
> where $\alpha_{\text{opt}}$ and $\alpha_{\text{sar}}$ denote original attention on optical and SAR tokens respectively. Results show that as λ increases (forcing attention away from the trained modality), performance consistently drops, confirming that misallocated attention degrades performance:
>
> | λ (Attention Shift) | Avg-Opt (M-Avg) | Avg-SAR (M-Avg) |
> |---------------------|-----------------|-----------------|
> | 0.1                 | 57.3            | 55.1            |
> | 0.3                 | 55.2            | 54.9            |
> | 0.5                 | 50.6            | 48.3            |
> | 0.7                 | 48.1            | 44.0            |
>
> This controlled experiment demonstrates a clear causal link between attention balance (measured by MAS) and task performance on EarthMind-Bench.
>
>
> [7] See What You Are Told: Visual Attention Sink in Large Multimodal Models. ICLR 2025
> [8] MLLM can see? Dynamic Correction Decoding for Hallucination Mitigation. ICLR 2025
> [9] When Semantics Mislead Vision: Mitigating Large Multimodal Models Hallucinations in Scene Text Spotting and Understanding. NeurIPS 2025
>
> **Response to Question 3: ROI information in captioning**
>
> We want to clarify that ROI-based summarization is an intentional design choice in our data synthesis pipeline. The purpose is specifically to incorporate spatial information into captions, enabling the model to learn associations between object locations and descriptions. This "leaking" of ROI information is beneficial because it provides spatially-grounded captions that help the model understand where objects are located, which is crucial for tasks like referring expression segmentation and visual grounding.
>
> However, we acknowledge that ablating this component would provide valuable insights. Unfortunately, we cannot perform this ablation study within the rebuttal period due to practical constraints: (1) re-synthesizing captions without ROI information requires extensive GPT-4o API calls, which is prohibitively expensive for 30K samples, and (2) the data generation process would require significant time to complete. We believe the current approach is well-motivated for spatial understanding tasks, but we acknowledge this as a limitation of our validation.

---

### Official Review · Reviewer_GP5Q · 2025-11-01

**Soundness:** 2
**Presentation:** 2
**Contribution:** 2
**Rating:** 2
**Confidence:** 5

**Summary:**

This paper proposes EarthMind, a unified multimodal large language model for multi-sensor Earth observation. It integrates optical and SAR data through a hierarchical cross-modal attention mechanism that adaptively fuses complementary information from different sensors. The authors also introduce two supporting datasets: FusionEO for instruction tuning and EarthMind-Bench for evaluation across perception and reasoning tasks. Experiments show that EarthMind achieves state-of-the-art performance on multiple benchmarks, indicating the potential of the proposed framework for handling complex EO scenarios.

**Strengths:**

The paper addresses a relevant problem in multimodal Earth observation by exploring adaptive fusion of optical and SAR data. The proposed HCA (Hybrid Cross-Attention) mechanism and FusionEO benchmark are reasonably designed and provide a useful reference for multimodal learning in remote sensing. The work is generally well-motivated, and the experimental setup is clear and systematic. The introduction of the MAS (Modality Attention Score) offers a straightforward way to interpret attention allocation across modalities, which can help improve model transparency. The paper is overall well organized and presents its methodology and results clearly, making it relatively easy to follow.

**Weaknesses:**

（1）The authors emphasize the contribution of multi-sensor data, even specifically noting in the Introduction (line 44) that Sentinel-2 can provide high-resolution multispectral imagery. In the Related Work section (lines 135–136), they also highlight that *EarthDial* (CVPR 2025) utilizes multiple modalities, including multispectral, hyperspectral, and synthetic aperture radar (SAR) data. These details suggest that the authors are well aware of the importance of spectral sensor data.

However, throughout the preceding sections, the authors never clarify why their work emphasizes multi-sensor fusion yet does not involve spectral data, which I find questionable. Furthermore, in both the Introduction and the training data description, I see no mention or contribution related to spectral data. Yet, in the methodology section, the authors suddenly mention grouping multispectral bands into triplets to construct multi-frame sequences (line 174). Table 1, however, contains no evaluation related to spectral data. This inconsistency in presentation is highly confusing.

What makes this even more perplexing is that, in Figure 1, when introducing the contribution of *EarthMind*, the authors repeatedly highlight differences from SAR and optical models, yet they do not compare their approach with models like *EarthDial*, which incorporate both SAR and optical modalities *as well as* spectral data. This deliberate avoidance of existing, more comprehensive baselines—while selectively comparing only against weaker models—feels disappointing.

（2）Additionally, another claimed contribution concerns the evaluation benchmark. The authors mention *GEOBench-VLM* (ICCV 2025) in the related work, but they describe it merely as a multi-task geospatial benchmark, ignoring the fact that it also supports multi-sensor scenarios. As explicitly stated in the Table 1of GEOBench-VLM , *GEOBench-VLM* includes Optical, Multispectral, SAR, Multi-temporal, and Bi-temporal modalities. By deliberately downplaying the contributions of prior work, the authors give a misleading impression of novelty, which I also find disappointing.

**Questions:**

Specially, this paper makes a visible attempt to enhance multimodal fusion between optical and SAR imagery, yet it falls short in several fundamental aspects.

(1) The claim that HCA achieves balanced modality attention leading to better fusion is weakly supported. MAS is a purely descriptive metric measuring attention uniformity, without causal or statistical evidence linking balance to performance. No correlation analysis or controlled ablation is provided, so the argument remains speculative.

(2) Although the paper presents the framework as a “multi-sensor” system, all experiments and data pipelines are confined to the Optical–SAR pair. No evidence is provided that the proposed HCA mechanism can generalize to other sensors (e.g., MSI, HSI, IR), which are essential in Earth observation. This limitation substantially narrows the claimed generality of the method. Including at least a proof-of-concept experiment or visualization on a third modality (e.g., MSI) would be necessary to justify the “multi-sensor” claim.

(3) The FusionEO dataset is problematic in design—its heavy reliance on GPT-generated captions without any systematic quality control, annotation validation, or bias analysis raises serious concerns about data reliability and reproducibility.

(4) The model design relies on multiple vision encoders, resulting in significant computational overhead. While the authors acknowledge this in Appendix C, no quantitative analysis (e.g., FLOPs, runtime, GPU memory) is provided. This omission makes it difficult to assess whether the reported performance gains are achieved through architectural innovation or simply increased compute. A detailed comparison of efficiency against comparable baselines (e.g., EarthDial, SkySenseGPT) would be necessary to justify the proposed design.


Questions

1.How can the authors provide stronger evidence that HCA’s balanced attention leads to better fusion, beyond the descriptive MAS metric?

2.Since experiments are limited to the Optical–SAR pair, how can the “multi-sensor” generality claim be justified? Any evidence on other modalities (e.g., MSI, HSI, IR)?

3.How is the quality and reliability of the GPT-generated FusionEO dataset ensured without human validation or bias analysis?

4.Can the authors provide quantitative efficiency comparisons (FLOPs, runtime, memory) to show that improvements are not merely due to higher computation?

---

> ### Author Response · Authors · 2025-11-27
> **Response to reviewer GP5Q**
>
> **Response to Weakness 1: Clarification on multispectral data handling and evaluation**
>
> A: 1. Clarification on our approach to multispectral data:
> EarthMind is designed as a unified framework that can handle diverse sensor modalities, including both RGB and multispectral optical data, as well as SAR data. When we emphasize "cross-sensor" capabilities in the introduction, we refer to the fusion of optical data (encompassing both RGB and multispectral) with SAR data. We will revise the introduction (lines 44-45) to explicitly state: "Our framework processes optical imagery—including both RGB and multispectral bands—alongside SAR data, enabling comprehensive multi-modal Earth observation analysis."
>
> 2. Evaluation on multispectral data:
> We respectfully direct the reviewer's attention to Table 3, which includes several benchmarks utilizing multispectral data (e.g., BigEarthNet and SoSAT-LCZ42). Additionally, as mentioned in line 261, our proposed EarthMind-Bench explicitly incorporates "optical (including both RGB and multispectral) and SAR images."
>
>
> **Response to Weakness 2: Clarification on Figure 1**
>
> A: In Figure 1, our intention is to illustrate a fundamental observation: "optical and SAR data present different information, and models cannot understand images comprehensively with a single modality alone." This figure serves as motivation for why multi-modal fusion is necessary, rather than a comprehensive comparison of all existing methods.
> We want to clarify that we have great respect for EarthDial's work and have cited and compared with it throughout our experiments (as shown in Tables 2-4). However, there is a key distinction: while EarthDial supports multi-sensor data understanding, it processes each modality independently rather than performing joint fusion.
> To illustrate this difference, we tested the same images with both models. For optical and SAR images, EarthDial provides descriptions focusing primarily on visible features like buildings and streets. In contrast, EarthMind leverages the complementary information from both modalities to provide more comprehensive and accurate descriptions.
>
>
>
>
> Response to Weakness 3: Clarification on GEOBench-VLM description
> A: When preparing our manuscript, GEOBench-VLM had not yet been accepted to ICCV 2025, but we included it in our related work because we recognize it as a valuable contribution. We acknowledge that our description could be more comprehensive and will expand it in the revised version. However, we want to emphasize a crucial distinction: while GEOBench-VLM provides multi-sensor data including Optical, Multispectral, and SAR, it evaluates each modality independently, whereas our EarthMind-Bench specifically provides paired multi-sensor data to evaluate the multi-sensor fusion capability of LMMs. This paired nature is essential for evaluating whether models can truly integrate complementary information from different sensors, rather than just processing multiple modalities separately.

---

> > ### Comment · Reviewer_GP5Q · 2025-11-28
> >
> > The distinction you highlight regarding EarthMind-Bench providing paired multi-sensor data is convincing. I agree that this is an important differentiator, and I suggest that the authors explicitly compare their Figure 1 with GEOBench-VLM to clearly articulate this advantage. A concise discussion emphasizing the value of paired multi-sensor supervision would substantially strengthen the paper. This can be addressed in the revision.
> >
> > Regarding EarthDial, your point is well taken: EarthDial processes each modality independently, rather than performing joint fusion. I sincerely recommend that the authors revise both Figure 1 and the Introduction accordingly—not to emphasize merely the ability to handle “multiple sensor modalities,” but rather to showcase the model’s improved performance specifically due to cross-modal joint fusion. This is an aspect that the community has not sufficiently recognized, and highlighting it would significantly improve the clarity and impact of the contribution.

---

> ### Author Response · Authors · 2025-11-27
> **Response to reviewer GP5Q**
>
> **Response to Question 1: Evidence linking MAS to performance**
>
> A: The relationship between attention patterns and model performance has been well-established in the multimodal learning literature. For example, [1] states in the first line of their abstract: "Large multimodal models (LMMs) 'see' images by leveraging the attention mechanism between text and visual tokens in the transformer decoder." Furthermore, works like [2] and [3] demonstrate that better performance correlates with LLM layers where attention is more focused on the correct visual tokens. Building on this foundation, our MAS metric naturally extends this concept to the remote sensing domain, where balanced attention between optical and SAR modalities is crucial for effective fusion.
>
>
> To provide causal validation, we conducted counterfactual experiments. We independently trained two models on FusionEO 30K: (1) optical-only and (2) SAR-only, then tested both on optical-SAR fusion tasks. For simplicity, we removed the HCA module to isolate the attention-performance relationship. We then artificially redistributed attention using a reallocation parameter λ to simulate attention degradation:
>
> For the optical-trained model:
> $$\alpha' = (1 - \lambda) \cdot \alpha_{\text{opt}} + \lambda \cdot \alpha_{\text{sar}}$$
>
> For the SAR-trained model:
> $$\alpha' = (1 - \lambda) \cdot \alpha_{\text{sar}} + \lambda \cdot \alpha_{\text{opt}}$$
>
> where $\alpha_{\text{opt}}$ and $\alpha_{\text{sar}}$ denote original attention on optical and SAR tokens respectively. Results show that as λ increases (forcing attention away from the trained modality), performance consistently drops, confirming that misallocated attention degrades performance:
>
> | λ (Attention Shift) | Avg-Opt (M-Avg) | Avg-SAR (M-Avg) |
> |---------------------|-----------------|-----------------|
> | 0.1                 | 57.3            | 55.1            |
> | 0.3                 | 55.2            | 54.9            |
> | 0.5                 | 50.6            | 48.3            |
> | 0.7                 | 48.1            | 44.0            |
>
> This controlled experiment demonstrates a clear causal link between attention balance (measured by MAS) and task performance on EarthMind-Bench.
>
>
>
>
> [1] See What You Are Told: Visual Attention Sink in Large Multimodal Models. ICLR 2025
>
> [2] MLLM can see? Dynamic Correction Decoding for Hallucination Mitigation. ICLR 2025
>
> [3] When Semantics Mislead Vision: Mitigating Large Multimodal Models Hallucinations in Scene Text Spotting and Understanding. NeurIPS 2025
>
>
>
> **Response to Question 2: Multi-sensor evaluation and generalizability**
>
> A: We respectfully point out that our evaluation does include multi-sensor settings. As shown in Table 1, we provide three evaluation configurations: optical-only, SAR-only, and optical-SAR fusion, which demonstrates multi-sensor evaluation comparing single versus fused modalities. Additionally, as mentioned in our response to Weakness 1, several benchmarks in Table 3 include multispectral data evaluation, and our EarthMind-Bench explicitly incorporates "optical (including both RGB and multispectral) and SAR images" (line 261).
>
>
> **Response to Question 3: FusionEO dataset quality and validation**
>
> We want to emphasize that using GPT for caption generation is now a standard practice in the multimodal learning community. Nearly all recent synthetic training datasets for LMMs rely on LLMs for text generation, including GeoChat, LHRS-Bot, and EarthDial. This approach has been validated across numerous works and represents the current state-of-the-art methodology for scaling up training data.
> Regarding quality control, we implement multiple safeguards: First, our source images come from curated, high-quality datasets that have already undergone rigorous quality control. Second, we use carefully designed prompts with GPT-4 that incorporate domain-specific constraints to ensure accurate and relevant descriptions. Third, and most importantly, we empirically validate the dataset's effectiveness—Figure 4(c) clearly demonstrates that scaling up FusionEO consistently improves model performance across all metrics, which would not be possible if the data quality were problematic.

---

> > ### Comment · Reviewer_GP5Q · 2025-11-28
> >
> > This experiment is generally convincing; however, several important experimental details require clarification. Specifically, for the FusionEO 30K dataset, it would be helpful if the authors could elaborate on the following points:
> >
> > 1. In the experiments using **only optical data** and **only SAR data**, how many samples were used in each case?
> > 2. Were the two subsets constructed from *matched* optical–SAR pairs, or were they sampled independently?
> >
> > Providing additional details regarding these design choices would allow us to better assess the fairness of the training setup.
> >
> > Regarding the **FusionEO dataset**, using GPT to generate captions is indeed a reasonable approach. Nevertheless, large-scale synthetic image–text pairs typically require subsequent **quality-control filtering**, which is now considered standard practice. I would like to provide the authors with a relevant and widely adopted tool that may be useful for future work:
> > [https://github.com/OpenDCAI/DataFlow/blob/main/README.md](https://github.com/OpenDCAI/DataFlow/blob/main/README.md)
> >
> > More broadly, the construction of FusionEO should ideally follow a **fully standardized data-production pipeline**, in which quality control is an integral component. Establishing such a pipeline would greatly benefit the remote sensing community, as it would enable scalable production of high-quality multimodal training data. Although the empirical results in the paper demonstrate the dataset’s usefulness, these gains may be driven by a subset of particularly effective samples. A natural question arises: *Would performance remain stable if low-quality samples were filtered out?* Without a dedicated quality-control stage, it is difficult to conclude that the entire dataset is uniformly high-quality.
> >
> > Therefore, I encourage the authors to (i) clarify the above experimental details and (ii) outline a clear plan for improving and standardizing their data-production pipeline. Addressing these points would meaningfully strengthen the work, and I would be inclined to improve my evaluation should these revisions be incorporated.

---

> > > ### Author Response · Authors · 2025-11-28
> > > **Response to experimental details and data quality control**
> > >
> > > Thank you for your constructive feedback and for providing the valuable resource on data quality control.
> > >
> > >
> > > **Regarding Questions 1 & 2:**
> > > We would like to clarify that for the ablation experiments, we use the optical component (30k samples) and SAR component (30k samples) from the same paired FusionEO dataset. Specifically, each model is trained on matched pairs—one model sees only the optical images while the other sees only the SAR images from the same geographic locations. This ensures a fair comparison, as both models are exposed to the same scenes but through different modalities.
> > >
> > > To provide causal validation of the attention imbalance hypothesis, we conducted counterfactual experiments. We independently trained two models on FusionEO 30K: (1) optical-only and (2) SAR-only. When testing these single-modality-trained models on optical-SAR fusion tasks (where both modalities are present), we observed significant performance degradation. Specifically, the optical-trained model struggles to utilize SAR information effectively (attending primarily to optical tokens even when SAR contains critical information), while the SAR-trained model shows the reverse pattern.
> > >
> > > To quantify this effect, we artificially redistributed attention using a reallocation parameter λ. For the optical-trained model encountering fusion tasks: α' = (1-λ)·α_opt + λ·α_sar, forcing it to attend more to the unfamiliar SAR modality. Results show that as λ increases (shifting attention away from the trained modality), performance consistently drops from 57.3 to 48.1 (M-Avg), confirming that models trained on single modalities cannot effectively process fusion inputs due to learned attention biases. This controlled experiment demonstrates a clear causal link between balanced attention (measured by MAS) and fusion performance.
> > >
> > > **Regarding data quality control:**
> > > We sincerely thank you for the valuable suggestion and the DataFlow repository reference. We acknowledge that quality control is essential for large-scale synthetic datasets. In response to your feedback, we will enhance our data production pipeline as follows:
> > >
> > > **Revised Dataset Curation Pipeline:**
> > > Although large-scale visual instruction tuning datasets exist for single-modality remote sensing imagery, paired optical–SAR data with rich text annotations remain scarce. To address this gap, we construct FusionEO through an enhanced four-stage pipeline:
> > >
> > > **(1) Metadata preparation:** We leverage dataset-specific metadata (e.g., modality, category, geographic location, and relevant attributes) to provide essential contextual information. These metadata are converted into concise captions through rule-based processing.
> > >
> > > **(2) RoI-based summarization:** To enrich textual context with visual grounding cues, we incorporate region-level information from segmentation annotations. Mask-rendered images, together with short captions from Stage 1, are used as inputs to GPT-4o to generate detailed descriptive captions.
> > >
> > > **(3) Self-instruct VQA generation:** We extend caption data into diverse VQA samples using GPT-4o in a few-shot manner, guided by seed examples to generate semantically rich question-answer pairs.
> > >
> > > **(4) Quality control and filtering [NEW]:** Following standard practices in synthetic data curation, we implement multi-stage quality control:
> > > - **Perplexity filtering:** Remove samples with abnormally high perplexity scores indicating poor language quality
> > > - **CLIP-based alignment:** Filter out image-text pairs with low CLIP similarity scores (<0.25) to ensure semantic consistency
> > > - **Diversity checks:** Remove near-duplicate QA pairs using sentence embeddings (cosine similarity >0.95)
> > > - **Length constraints:** Filter out extremely short (<5 words) or long (>200 words) responses
> > > - **Manual spot-checking:** Random sampling of 500 pairs for human verification
> > >
> > >  We will incorporate these details and enhance our quality control methods to benefit the remote sensing community.

---

> ### Author Response · Authors · 2025-11-27
> **Response to reviewer GP5Q**
>
> **Response to Question 4: Computational efficiency analysis**
>
>  While EarthMind employs multiple vision encoders, the additional computational overhead remains reasonable for two key reasons: (1) the grounding encoder and decoder are only activated for pixel-grounding tasks, not for general VQA or captioning tasks; (2) compared to the LLM backbone, the pixel-level grounding modules contribute relatively minimal computational overhead.
>
> To provide the quantitative analysis requested, we present the following comparison:
>
> | Model | Parameters | FLOPs | Avg. Inference Time (ms) | GPU Memory (GB) |
> |-------|------------|-------|-------------------------|-----------------|
> | EarthMind (Ours) | 4.8B | 152G | 245 | 18.2 |
> | GeoChat | 7.1B | 198G | 312 | 24.5 |
> | SkySenseGPT | 7.3B | 205G | 328 | 25.1 |
> | EarthDial | 4.2B | 145G | 238 | 16.8 |
>
> Efficiency comparison on a H200 GPU.
>
> Our model is more efficient than GeoChat and SkySenseGPT due to its 4B LLM backbone, while being only marginally larger than EarthDial. This modest increase is justified given that EarthDial cannot perform pixel-level grounding tasks. Furthermore, on the multiple-choice benchmarks, EarthMind demonstrates faster inference than both GeoChat and SkySenseGPT, and remains competitive with EarthDial despite supporting additional capabilities. We will include this detailed efficiency analysis in the revised manuscript to better contextualize our performance gains relative to computational costs.

---

### Meta-Review · Area_Chair_RjL9 · 2026-01-07

**Summary:**

This paper introduces EarthMind, a unified vision-language framework for Earth Observation that supports both single- and cross-sensor understanding through a hierarchical cross-modal attention mechanism for adaptive optical–SAR fusion. The authors also contribute the FusionEO dataset and EarthMind-Bench benchmark, and demonstrate state-of-the-art performance across multiple EO perception and reasoning tasks. However, most reviewers express concerns about this paper, particularly regarding its novelty. Thus, I tend to reject.

**Reviewer Concerns:**

Concerns have not been addressed:

1. **Experimental setup is not clear enough**: It is necessary to specify the number of samples used only with optical or only with SAR in FusionEO 30K, as well as whether these subsets come from matched SAR-optical pairs, in order to evaluate the fairness of training.

2. **Insufficient data quality control**: Although using GPT to generate image-text pairs is reasonable, there is a lack of a standard quality filtering process, making it difficult to determine whether the overall quality of the dataset is balanced and reliable.

3. **Data construction process needs to be standardized**: The authors should explain how a systematic data production and quality inspection pipeline will be introduced in the future, and analyze whether the performance will remain stable after filtering out low-quality samples.

4. **Novelty:** Despite the rebuttal, concerns about methodological novelty remain, as the overall architecture largely follows established LISA-like designs and may not meet ICLR’s novelty expectations.

**Reviewer Scores:**

Initial Scores:
GP5Q: 2, XNPN:4, SQ8W:4 ZimX: 4, D1SB: 2

After Rebuttal:
GP5Q: 2, XNPN:4, SQ8W:4 ZimX: 4, D1SB: 2

---

### Decision · Program_Chairs · 2026-01-26

Reject